# Structural model for differential cap maturation at growing microtubule ends

**Juan Estévez-Gallego[1†], Fernando Josa-Prado[1†], Siou Ku[2†], Ruben M Buey[1,3], Francisco A Balaguer[1], Andrea E Prota[4], Daniel Lucena-Agell[1], Christina Kamma-Lorger[5], Toshiki Yagi[6], Hiroyuki Iwamoto[7], Laurence Duchesne[2], Isabel Barasoain[1], Michel O Steinmetz[4,8], Denis Chrétien[2], Shinji Kamimura[9], J Fernando Díaz[1], Maria A Oliva[1]\***

[1]Structural and Chemical Biology Department, Centro de Investigaciones Biológicas, CSIC, Madrid, Spain; [2]Univ Rennes, CNRS, IGDR (Institut de Génétique et Développement de Rennes) – UMR 6290, Rennes, France; [3]Departamento de Microbiología y Genética, Universidad de Salamanca-Campus Miguel de Unamuno, Salamanca, Spain; [4]Division of Biology and Chemistry, Laboratory of Biomolecular Research, Paul Scherrer Institut, Villigen, Switzerland; [5]ALBA synchrotron, CELLS, Cerdanyola del Vallès, Spain; [6]Department of Life Sciences, Faculty of Life and Environmental Sciences, Prefectural University of Hiroshima, Hiroshima, Japan; [7]Diffraction and Scattering Division, Japan Synchrotron Radiation Research Institute, Hyogo, Japan; [8]University of Basel, Biozentrum, Basel, Switzerland; [9]Department of Biological Sciences, Faculty of Science and Engineering, Chuo University, Tokyo, Japan

**\*For correspondence:**
marian@cib.csic.es

[†]These authors contributed equally to this work

**Competing interests:** The authors declare that no competing interests exist.

**Abstract** Microtubules (MTs) are hollow cylinders made of tubulin, a GTPase responsible for essential functions during cell growth and division, and thus, key target for anti-tumor drugs. In MTs, GTP hydrolysis triggers structural changes in the lattice, which are responsible for interaction with regulatory factors. The stabilizing GTP-cap is a hallmark of MTs and the mechanism of the chemical-structural link between the GTP hydrolysis site and the MT lattice is a matter of debate. We have analyzed the structure of tubulin and MTs assembled in the presence of fluoride salts that mimic the GTP-bound and GDP•$P_i$ transition states. Our results challenge current models because tubulin does not change axial length upon GTP hydrolysis. Moreover, analysis of the structure of MTs assembled in the presence of several nucleotide analogues and of taxol allows us to propose that previously described lattice expansion could be a post-hydrolysis stage involved in $P_i$ release.

## Introduction

Microtubules (MTs) are ubiquitous cytoskeletal polymers built from α/β-tubulin heterodimers that assemble into a pseudo-helical cylinder. They are responsible for essential processes during cell growth and division, including chromosome segregation, intracellular transport, cell support and motility (*Desai and Mitchison, 1997*). MTs perform these functions by serving as scaffolds for other proteins and engaging mechanical forces through their dynamic behavior (*Gigant et al., 2000*; *Koshland et al., 1988*). Due to its central role in cell biology, tubulin is a reference target for antitumor drugs that modulate protein dynamics. Therefore, understanding the molecular mechanisms of tubulin activation and deactivation is crucial to designing more effective compounds that overcome cell resistance and lower the toxicity of compounds in clinical use.

Tubulin exists in two different conformations related to its polymerization state: curved (depolymerized) and straight (assembled into MTs) (*Buey et al., 2006*; *Gigant et al., 2000*;

*Nawrotek et al., 2011*; *Rice et al., 2008*). The binding and hydrolysis of guanosine nucleotides rule the polymerization-depolymerization of tubulin through chemical-linked conformational stages. GDP-tubulin remains inactive in the cytoplasm and the GTP exchange at the exchangeable site (E) on β-tubulin activates the α/β-heterodimers, providing interacting surfaces prone to the addition onto growing bent sheets or protofilaments (PFs) at the MT end (*Chrétien et al., 1995*; *McIntosh et al., 2018*). The formation of lateral contacts between PFs at the MT tip contributes to tubulin straightening (*Nogales and Wang, 2006*), which is key to creating a hydrolysis competent state (*Nogales et al., 1998*; *Oliva et al., 2004*). GTP hydrolysis at the E-site in β-tubulin induces conformational changes (*Alushin et al., 2014*) driving the 'peeling-off' disassembly of MTs (*Chrétien et al., 1995*; *Mandelkow et al., 1991*). MTs' continuous growth and shrinkage generate the motion of these filaments, which is known as dynamic instability (*Mitchison and Kirschner, 1984a*) and involves transient polymer intermediates adopting various nucleotide and conformational states, of which we are getting the first structural glimpses (*Alushin et al., 2014*; *Zhang et al., 2015*; *Zhang et al., 2018*). The GDP-bound tubulin forming the body of the MT has a compact, straight and regular lattice, whereas the tip of the MT contains GTP- and GDP•$P_i$-bound tubulin molecules, and is known as the GTP-cap. This growing MT end varies in size (*Duellberg et al., 2016b*) and decreases in stability with age due to GTP hydrolysis and/or $P_i$ release (*Carlier et al., 1984*; *Duellberg et al., 2016b*; *Duellberg et al., 2016a*; *Gardner et al., 2011*; *Mitchison and Kirschner, 1984a*; *Padinhateeri et al., 2012*). The GTP-cap prevents MT depolymerization but its lattice pattern is poorly understood. It has been described as having a tapered shape (*Mandelkow et al., 1991*) or as outwardly curved sheets (*Atheton et al., 2018*; *Chrétien et al., 1995*; *Guesdon et al., 2016*), though a recent study suggests flared, curved PFs at growing MT ends (*McIntosh et al., 2018*). It is believed that when GTPase activity reaches the tip and no new capping tubulins are added, MTs switch from growing to shrinking in a multi-step process (*Schek et al., 2007*; *Walker et al., 1991*) required for the GTP-cap to disappear.

Insight into the nature of the GTP-cap is fundamental to understanding the mechanisms governing MT dynamics and developing new modulating compounds targeting MTs. Here, we address this question using a combination of high- and low-resolution structural techniques with a biochemically controlled in vitro system applied to tubulin and MTs. The use of multiple γ-phosphate analogues (BeF$_3^-$, AlF$_x$) and nucleotides (GMPCPP, GMPPCP, GMPCP) allows us to develop systems that model MTs in their GTP-bound, transitional (GDP•$P_i$) and metastable (GDP-bound) states. In this model, all tubulin states would be compacted, which contrast with the widely accepted model of an expanded GTP-lattice that would compact after GTP hydrolysis (*Alushin et al., 2014*). We further propose that if previously observed lattice expansion actually occurs during cap maturation, it happens at a post-hydrolysis stage, between the transitional (GDP•$P_i$) and the metastable GDP-bound states, being an expanded intermediate conformational stage required for $P_i$ release.

## Results

### Phosphate analogues mimic activation and transition states at the hydrolytic E-site

AlF$_x$ and BeF$_x$ are small inorganic molecules that mimic the chemical structure of phosphate (*Bigay et al., 1987*) and can bind and activate GDP-bound proteins (*Díaz et al., 1997*; *Mittal et al., 1996*). BeF$_x$ complexes are strictly tetrahedral due to the sp3 orbital hybridization, whereas AlF$_x$ is hexacoordinate (*Coleman et al., 1994*; *Martin, 1988*) and closely resembles the bipyramidal transition state of phosphate. We combined structural and biochemical studies to validate and correlate the complexes of GDP-tubulin with these γ-phosphate analogues, in both GTP-activated (BeF$_3^-$) and GDP•$P_i$ transition (AlF$_4^-$/AlF$_3$) states.

To determine the crystal structure of GDP-tubulin bound to each of the γ-phosphate analogues, we used a tubulin complex with RB3 and TTL (*Figure 1A*), referred to T$_2$R-TTL (*Prota et al., 2013a*). We included a GDP exchange step during sample preparation (*Díaz and Andreu, 1993*) to ensure the GDP's presence at the E-site. Before setting crystallization plates, we incubated the GDP-T$_2$R-TTL complexes with either BeF$_3^-$ or AlF$_x$ (prepared in situ from mixtures of AlCl$_3$ and HKF$_2$ to avoid Al precipitation) to produce GDP-T$_2$R-TTL-BeF$_3^-$ and GDP-T$_2$R-TTL-AlF$_x$ complexes, respectively. We noticed that BeF$_3^-$ and AlF$_x$ cleared easily from the nucleotide binding pocket during crystals

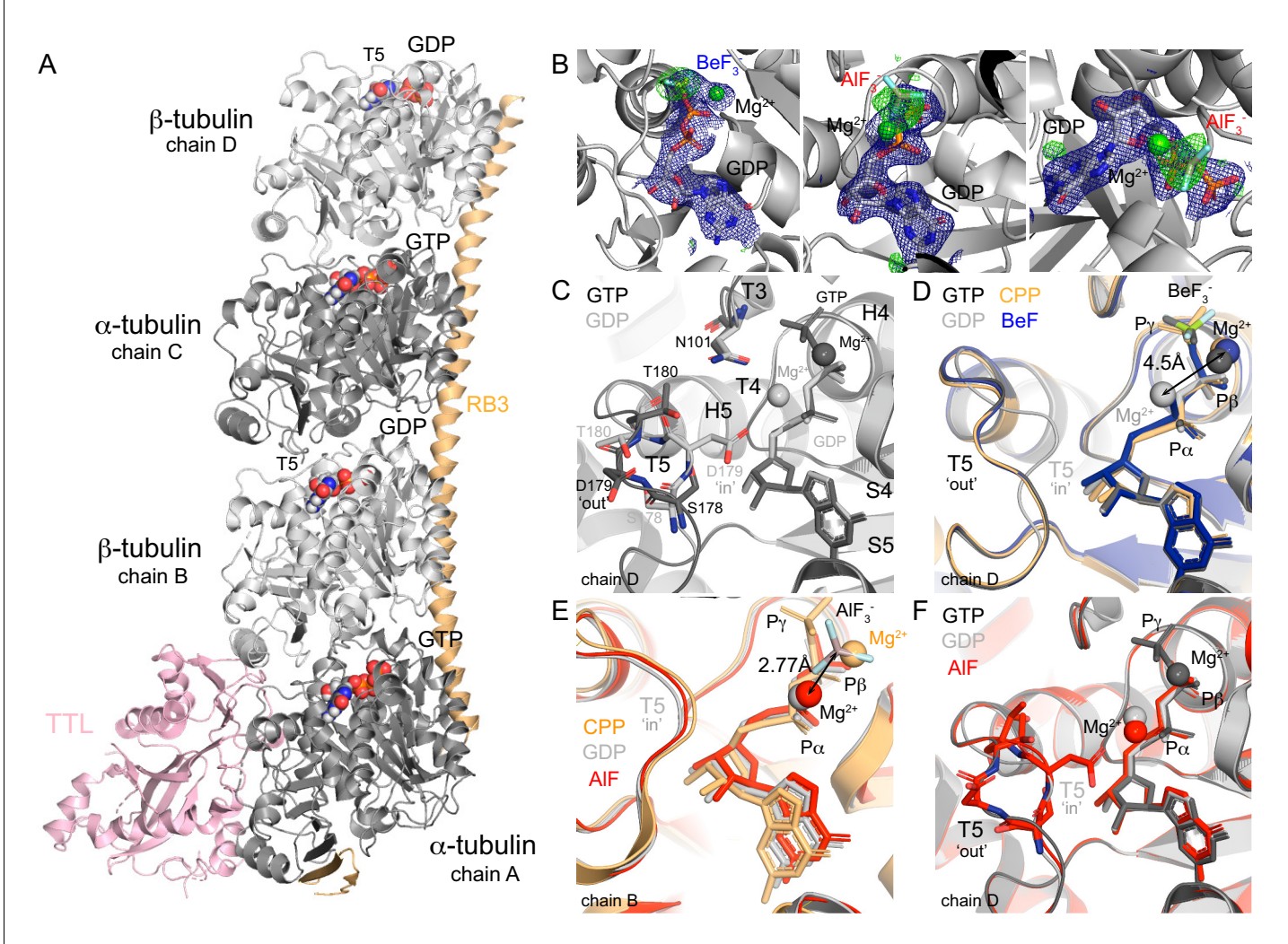

**Figure 1.** Structure of tubulin bound to GDP-phosphate analogues. (**A**) The $T_2R$-TTL complex includes one RB3 molecule (orange), one TTL molecule (pink) and two tubulin heterodimers: α-tubulin (dark gray, GTP-bound, chains A and C), β-tubulin (light gray, GDP-bound, chains B and D). (**B**) Composite omits maps of $BeF_3^-$ (*left*) and $AlF_3$ (*middle* and *right*): mFo-DFc maps (blue, contour level 1.0) of nucleotide and $Mg^{2+}$ ions combined with 2mFo-DFc maps (green, contour level 3.0) of the related phosphate analogues. (**C**) GTP (PDB 5xp3, black) and GDP- (PDB 4i55, gray) bound β-tubulin highlighting the localization of secondary structural elements surrounding the nucleotide-binding site according to *Löwe et al. (2001)* and alternative positions of T5 loop depending on the nucleotide-bound state. (**D**) Chain D alignment of GDP-$BeF_3^-$ structure (blue) with GMPCPP-bound (PDB 3ryh, orange), GTP-bound (black) and GDP-bound (gray) structures showing $BeF_3^-$/γ-phosphates co-localization, differences on the position of $Mg^{2+}$ ions depending on the nucleotide-bound state (4.5 Å apart), and T5 loop in a GTP-like ('out') conformation. (**E**) Chain B alignment of GDP-$AlF_3$ (red) structure with GMPCPP-bound (orange) and GDP-bound (gray) structures highlighting that $AlF_3$ sits out of the γ-phosphate site at 2.77 Å from the $Mg^{2+}$ ion (in a position equivalent to that on the GDP-bound structure). (**F**) Chain D alignment of GDP-$AlF_3$ (red) structure with GTP-bound (black) and GDP-bound (gray) structures showing the dual conformation of T5 loops.

manipulation, which could be related to their low binding affinity to soluble curved tubulin (see below). After several attempts, we found the $BeF_3^-$ and $AlF_3$ moieties at the E-site of chains D and B (*Figure 1A–B*), respectively.

The GDP-$BeF_3^-$ structure (*Table 1*, data collection and refinement statistics), shows the phosphate analogue in a tetrahedral state at a distance from the β-phosphate that is similar to the distance observed between the β- and γ-phosphate atoms in the GTP-bound state (*Figure 1D*), and not further as expected for the transitional GDP•$P_i$ state (*Wittinghofer, 1997*). Consistent with reported GTP- and GMPCPP-bound tubulin structures, one $Mg^{2+}$ ion coordinates both the β-phosphate and the $BeF_3^-$ in a position clearly different than that of GDP-bound tubulin structures (~4.5 Å apart,

**Table 1.** Data collection and refinement statistics.

| | Native $T_2R$-TTL-$AlF_3$ (PDB 6s9e) | Native $T_2R$-TTL-$BeF_3^-$ (PDB 6gze) |
|---|---|---|
| **Data collection** | | |
| Space group | $P2_12_12_1$ | $P2_12_12_1$ |
| Cell dimensions | | |
| a, b, c (Å) | 104.999, 157.357, 180.261 | 104.176, 156.744, 180.587 |
| α, β, γ (°) | 90.00, 90.00, 90.00 | 90.00, 90.00, 90.00 |
| Resolution (Å) | 48.003–2.25 | 49.458–2.49 |
| $R_{merge}$ | 0.075 (1.222) | 0.071 (1.159) |
| $R_{pim}$ | 0.025 (0.417) | 0.028 (0.473) |
| I/σI | 16.5 (1.8) | 7.1 (0.6) |
| Completeness (%) | 99.0 (99.0) | 100 (100) |
| Redundancy | 9.6 (9.2) | 7.1 (7.0) |
| $CC_{half}$ | 0.979 (0.635) | 0.999 (0.993) |
| **Refinement** | | |
| Resolution (Å) | 48.003–2.25 | 49.458–2.49 |
| No. of reflections | 140102 | 103915 |
| $R_{work}/R_{free}$ | 0.2029/0.2278 | 0.2121/0.2565 |
| No. of atoms | 17701 | 16799 |
| Protein | 17279 | 16572 |
| Ligand | 223 | 175 |
| Water | 199 | 52 |
| B-factors | | |
| Protein | 64.0 | 80.4 |
| Ligand | 59.5 | 73.0 |
| Water | 45.7 | 67.5 |
| Wilson B | 48.90 | 64.70 |
| r.m.s deviation | | |
| Bond lengths (Å) | 0.002 | 0.003 |
| Bond angles (°) | 0.526 | 0.557 |
| Ramachandran % | | |
| Favor/allow/out | 97.88/2.12/0.00 | 97.52/2.48/0.00 |

[*]Data were collected from a single crystal.

[**]Values in parentheses are for the highest resolution shell.

*Figure 1D*). The GDP-$BeF_3^-$ complex is stabilized via hydrogen bonds and salt bridges contacts with loops T1, T4, T6 (GDP) and T3 and T4 ($BeF_3^-$, *Table 2*). Importantly, loop T5 is in a 'flipped-out' conformation, leaving D179 exposed to the solvent and putting T180 closer to N101 in loop T3 (*Figure 1C–D*) as shown previously in GTP-bound tubulin (*Nawrotek et al., 2011*). Thus, our results suggest that $BeF_3^-$ is a γ-phosphate analogue that mimics tubulin's GTP-bound state in the curved conformation, and not a GDP•$P_i$ or intermediate transition state of hydrolysis.

The $AlCl_3$/$HKF_2$ mixtures we used produce roughly ~50% $AlF_3$ and ~50% $AlF_4^-$ (*Goldstein, 1964*), and the related moiety was modeled as $AlF_3$ (*Table 1*, *Figure 1B*). Strikingly, this analogue did not occupy the position equivalent to the γ-phosphate as observed in other 'classic' GTPase structures of the GDP-$AlF_3$ complex (e.g. PDB 2ngr, 1grn, 2b92, 2g77, 4jvs and 4iru). Instead, we found the $AlF_3$ density beside the $Mg^{2+}$ ion (at a distance of 2.77 Å, *Figure 1B*), which was simultaneously coordinating α- and β-phosphates (i.e., GDP coordination, *Figure 1E*, comparison with $Mg^{2+}$

**Table 2.** PDBePISA analysis of nucleotide-hydrogen bonding at the E-site.

| | Curved conformation | | | | | Straight conformation | | | | |
|---|---|---|---|---|---|---|---|---|---|---|
| | GTP (5xp3) | GMPCPP (3ryh) | GDP (4i55) | BeF$_3^-$ (6gze) | AlF$_3$ (6s9e) | GMPCPP (3jat) | GMPCP (3jal) | GDP (3jar) | GTP-γ-S (3jak) | GDP•P$_i$ (6evx) |
| Base and ribose | Q11 S140 N206 N228 | S140 N228 | Q15 N206 N228 | N206 N228 | N206 N228 | N206 Y224 N228 | S140 N206 Y224 | Q15 S140 N206 Y224 | Q15 S140 N206 Y224 N228 | S140, N206 N228 |
| Pα | Q11 C12 | Q11 C12 S140 | C12 | C12 | C12 | Q11 C12 | Q11 C12 S140 | Q11 C12 | C12 | C12 |
| Pβ | Q11 G144 T145 G146 | Q11 T145 G146 | Q11 G144 T145 G146 | Q11 G144 T145 G146 | Q11 G144 T145 G146 | Q11 G144 T145 G146 | Q11 G144 T145 G146 | Q11 G144 T145 G146 | Q11 G144 T145 G146 | Q11 G144 T145 G146 |
| Pγ/ BeF$_3^-$/ AlF$_3$/ P$_i$/ | A99 G100 N101 G144 T145 | A99 G100 N101 G144 T145 | - | A99 G100 N101 T145 | E71 N101 Pα | A99 G100 G144 T145 | - | - | G144 T145 | T145 |
| Mg$^{2+}$ | yes | yes | yes | yes | yes | yes | no | no | no | no |

positioning of GMPCPP (orange) and GDP (gray) structures). AlF$_3$ was further stabilized through interactions with loops T2 and T3 and α-phosphate. We hypothesize that this configuration likely represents a transitional stage of P$_i$ release, where the Mg$^{2+}$ ion removes the γ-phosphate while moving to a new coordination position between α- and β-phosphates. Loop T5 in chain B is blocked due to tubulin axial contacts (*Figure 1A*); however, at chain D, this loop refined in a dual conformation during the structure model-building: 57% in GTP-like 'flip-out' and 43% in GDP-like 'flip-in' conformations (*Figure 1F*, comparison with GTP- (black) and GDP-bound (gray) structures). Since, we did not find any extra density at this chain denoting the presence of the analogue, we presume that very likely the AlF$_3$ or AlF$_4^-$ washed out and we captured loop T5 on its way back to the GDP-bound conformation.

## Phosphate analogues induce tubulin activation upon assembly into MTs

We analyzed the effect of γ-phosphate analogues on tubulin activation through time-course turbidity experiments in which we measured the assembly of MTs from fully substituted calf-brain GDP-tubulin in the presence of increasing BeF$_3^-$ or AlF$_x$ concentrations (37°C, no GTP added). Notice that our experiments were performed in MES buffer to avoid any competition effect of the commonly used phosphate buffer with the analogues (*Díaz and Andreu, 1993*). In this buffer condition, tubulin assembly requires glycerol under GTP (control experiments) or γ-phosphate analogues conditions, although GMPCPP does not require glycerol to induce assembly. In the presence of BeF$_3^-$ GDP-tubulin activation occurred at analogue concentrations above 1 mM (*Figure 2A*), suggesting a binding constant on the scale of mM. The polymerization curves also revealed that longer lag times occur with BeF$_3^-$ compared to GTP, indicating that nucleation is less efficient in the presence of GDP-BeF$_3^-$. This finding is also supported by the observation of fewer and longer MTs than in GTP control experiments (*Figure 2—figure supplement 1A*). However, we found that the equilibrium constant of addition of a tubulin dimer to the MT end is conserved as is shown by the fact that critical concentration (Cr) values (*Figure 2B*, 3.2 ± 0.4 µM vs. 2.9 ± 0.4 µM in GTP) remained similar across various BeF$_3^-$ concentrations. Also, increasing Mg$^{2+}$ concentration had a similar effect on the assembly under GDP-BeF$_3^-$ and GTP conditions, displaying similar slopes in the Wyman plot (0.37, *Figure 2C*). Thus, GDP-BeF$_3^-$ behaves like a GTP-bound state with slower MT kinetics. Likewise, AlF$_x$ induced tubulin activation with a binding constant in the mM range (*Figure 2D*, *Figure 2—figure supplement 1B*), which might reflect the fact that part of the molecules had their loop T5 in a GTP-bound conformation according to the crystal structure of AlF$_3$ tubulin complexes. Previous studies showed similar equilibrium binding constants of addition to the MT, regardless of AlF$_x$ concentration and the clear competition between both AlF$_x$ and BeF$_3^-$ for the same site on MTs (*Carlier et al., 1988*). These findings suggest that, (i) both γ-phosphate analogues likely display equivalent biochemical properties and; (ii) AlF$_x$ might occupy the γ-phosphate site on the straight tubulin conformation.

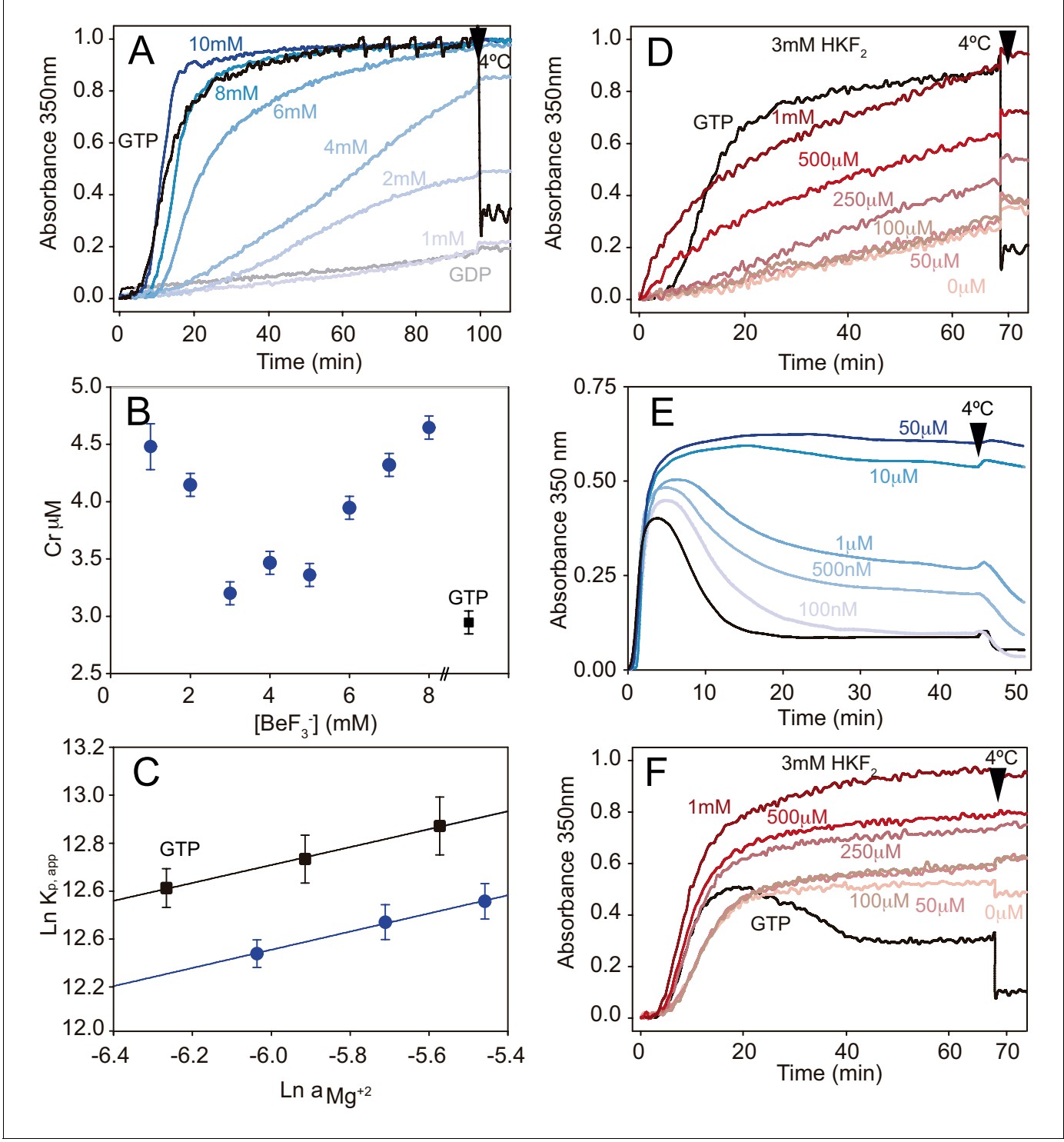

**Figure 2.** Phosphate analogues sustain tubulin activation and MT stabilization. (**A**) Time course assembly of 30 µM GDP-tubulin (gray line) with either 1 mM GTP (black line) or 1 mM GDP and increasing $BeF_3^-$ concentrations (1 mM, 2 mM, 4 mM, 6 mM, 8 mM and 10 mM; from light to dark blue). (**B**) Critical concentration (Cr) measurements of GDP-tubulin with 1 mM GTP (black square) or 1 mM, 2 mM, 3 mM, 4 mM, 5 mM, 6 mM, 7 mM and 8 mM $BeF_3^-$ (blue circles). (**C**) Wyman plots showing the effect of increasing $Mg^{2+}$ concentrations (4 mM, 5 mM, and 6 mM) on tubulin assembly (37˚C) in the presence of 1 mM GTP (black squares) or 1 mM GDP plus 3 mM $BeF_3^-$ (blue circles). Positive slopes indicate that at increasing $Mg^{2+}$ concentrations the ion incorporation into the filament is higher, and similar slope values (under $BeF_3^-$ and GTP) indicate a similar $Mg^{2+}$ dependency of the polymerization. (**D**) Time course assembly of 30 µM GDP-tubulin with either 1 mM GTP (black line) or 1 mM GDP, 3 mM $HKF_2$ and increasing concentrations of $AlCl_3$ (0 µM, 50 µM, 100 µM, 250 µM, 500 µM and 1 mM; from light to dark red). (**E**) Time course assembly of 30 µM tubulin in the absence of free GTP (black

*Figure 2 continued on next page*

Figure 2 continued

line) and in the presence of increasing concentrations of $BeF_3^-$ (100 nM, 500 nM, 1 µM, 10 µM and 50 µM; from light to dark blue) showing the stabilization effect of the analogue. (F) Time course assembly of 30 µM tubulin in the absence of free GTP and 0 mM $HKF_2$ (black line) or in the absence of free GTP with 3 mM $HKF_2$ and increasing concentrations of $AlCl_3$ (0 µM, 50 µM, 100 µM, 250 µM, 500 µM and 1 mM from light to dark red). Aluminum contamination in nucleotides and glass induce MT stabilization even when no $AlCl_3$ was added (light red). Arrows in graphs (A), (D), (E), and (F) indicate when samples were incubated at 4°C.

The online version of this article includes the following figure supplement(s) for figure 2:

**Figure supplement 1.** Electron micrographs of MTs polymerized in the presence of $BeF_3^-$ and $AlF_x$.

Finally, we verified the effect of these γ-phosphate analogues on the stabilization of MTs assembled in the presence of 30 µM GTP. $BeF_3^-$ inhibits MT disassembly at concentrations between 100–500 nM (*Figure 2E*) and hence, we estimate that the affinity of this γ-phosphate analogue for straight tubulin in assembled MTs (nM range, *Figure 2E*) is about three orders of magnitude higher than that for the curved GDP-bound, soluble protein (mM range, *Figure 2A*). These values correlate with those previously observed for Ras-like small GTPases where the binding affinity of $BeF_3^-$ is in the mM range (*Díaz et al., 1997*), but increases to the µM range with the addition of the GTPase Activating Protein (GAP, *Mittal et al., 1996*). Interestingly, MTs were far more stable at $BeF_3^-$ concentrations > 10 µM (i.e. the depolymerizing effect of non-physiological low temperatures disappeared, *Figure 2E*) likely because $BeF_3^-$ is retained at the γ-phosphate pocket, providing a stable MT conformation in the absence of $P_i$ release. When performing similar experiments to discern $AlF_x$ affinity to straight polymerized tubulin, we found that it is at least two orders of magnitude higher than this for the curved state because MTs were fully stabilized at the lower $AlCl_3$ concentration we used, 50 µM (*Figure 2F*) and tubulin activation upon assembly still required 0.5 mM $AlCl_3$ (*Figure 2D*). Unfortunately, $Al^{3+}$ contamination in the glass and nucleotides solutions (*Sternweis and Gilman, 1982*) prevented us from determining the strength of the interaction (*Figure 2F*).

## Phosphate analogues reveal lattice features for the GTP/GDP•$P_i$-bound states

We used X-ray fiber diffraction of aligned filaments in solution to determine the structural details of MTs in various nucleotide-bound states, which gave us the direct correlation between biochemical and structural data. We performed quick shear-flow alignment using methylcellulose (*Sugiyama et al., 2009*) at a constant, physiological temperature of 37°C, which allowed us to avoid temperature-related variations in axial repeat and filament diameter (*Kamimura et al., 2016*). We chose this technique because of the swiftness of getting results related to both the boundaries of the MT wall from the equatorial diffraction, and the axial repeats from the meridional diffraction (*Figure 3—figure supplement 1*; *Amos and Klug, 1974*; *Andreu et al., 1992*). Importantly, in these experiments, the signal-to-noise (S/N) ratio for the equatorial signals is very high because data includes the average of tens of millions of individual MTs. Therefore, we have estimated for the first time, the fraction of MT subpopulations according to their number of PFs (*Figure 3—figure supplement 1*, Materials and methods) using the $J_{04} + J_{N1}$ signals.

We first studied MTs polymerized in the presence of either 1 mM GTP (GDP-MTs, due to GTPase activity) or 1 mM GTP + 100 µM taxol (GDP-Tx-MTs) to analyze two known lattice conformations; compacted and expanded (*Alushin et al., 2014*; *Kellogg et al., 2017*). Under both experimental conditions, we found an axial tubulin repeat of 4 nm (*Figure 3A*, lines gray and brown and *Figure 3—figure supplement 2*). However, the 1 nm layer line, which is an harmonic of the 4 nm layer line, showed variations on their peaks distribution (*Figure 3A*, *inset*) indicating differences on the average monomer axial spacing between GDP-MTs and GDP-Tx-MTs. The existence of a second weaker set of ~8 nm layer lines on GDP-Tx-MTs further confirmed variations on the axial spacing between α- and β-tubulin. These experiments did not distinguish between intra-dimer (α-β) and inter-dimer (β-α) distances, but the averaged estimations of monomer lengths are similar to cryo-EM measurements (*Figure 4E*) and published cryo-EM structures (*Alushin et al., 2014*; *Kellogg et al., 2017*): 4.06 nm vs. 4.01 nm for GDP-MTs and 4.18 nm vs. 4.08 nm for GDP-Tx-MTs. Otherwise, the analysis of the equatorial diffraction showed similar diameters for GDP- and GDP-Tx-MTs (*Table 3*,

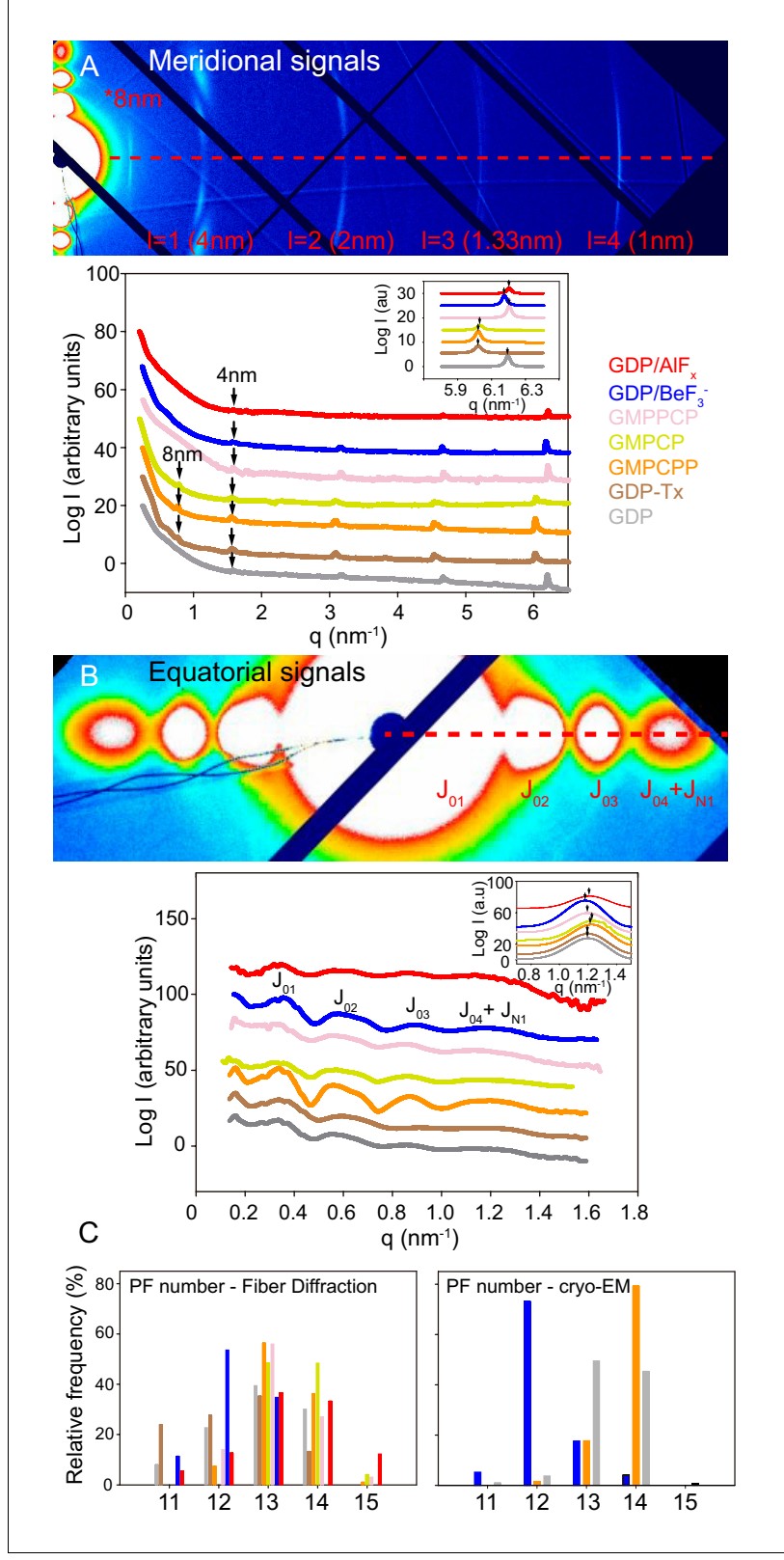

**Figure 3.** Fiber diffraction of MT models systems. GDP-BeF$_3^-$-MT (blue), GDP-AlF$_x$-MT (red), GMPPCP-MT (salmon), GMPCP-MT (yellow), GMPCPP-MT (orange), GDP-Tx-MT (brown) and GDP-MT (gray). (**A**) *Top*; representative image (GMPCPP-MTs) of meridional diffraction displaying the meridional plane from l = 1 (4nm) layer line and related harmonics (l = 2 to 4) for longitudinal metric calculations. *Bottom*; meridional intensity

*Figure 3 continued on next page*

*Figure 3 continued*

patterns, where arrows indicate the 4 nm and 8 nm peaks. The inset shows the best fit of 1 nm band experimental intensities in a Lorentzian normal distribution, highlighting positional differences between all tested MT growing conditions (peaks maxima, arrows). (**B**) *Top*; representative image (GMPCPP-MTs) of equatorial diffraction highlighting the equatorial plane (l = 0) for lateral metric calculations. *Bottom*; equatorial intensity patterns showing the corresponding Bessel functions from $J_{01}$ to $J_{04}+J_{N1}$. The inset shows the $J_{N1}$, calculated as described in M and M, displaying the differences in peak maxima (arrows) that occur under various nucleotide polymerization conditions. The red dash line on (**A**) and (**B**) top images shows planes used for intensity line plotting in $q_x$ space and further metric calculations. (**C**) Estimation of the number of PFs per MT and percentage of each subpopulation within the solution from fiber diffraction experiments (*left*) and cryo-EM images (*right*).

The online version of this article includes the following figure supplement(s) for figure 3:

**Figure supplement 1.** Shear-flow aligned fiber diffraction experiments.
**Figure supplement 2.** Shear-flow aligned fiber diffraction images.
**Figure supplement 3.** Shear-flow aligned fiber diffraction images of $BeF_3^-$- and $AlF_x$-MTs in the presence of taxol.

---

*Figure 3B*), although we calculated a slightly different fraction of MT subpopulations according to the number of PFs (*Figure 3C*). The PF number has no effect on the longitudinal spacing (*Alushin et al., 2014*; *Zhang et al., 2015*), but does affect the MT helical twist (*Chrétien and Wade, 1991*). Notice that in our fiber diffracting images the 4 nm layer line is slightly curved, which could be related to the super-twist existing in MTs with skewed PFs (*Chrétien and Fuller, 2000*). However, we relate this effect to a partial (2–3°) misalignment by shearing. Despite such differences, GDP- and GDP-Tx-MTs displayed very alike PFs lateral spacing (*Table 3*), supporting the equivalent lateral contacts found in the aforementioned cryo-EM structures.

Second, we measured the main features of MTs polymerized in the presence of $BeF_3^-$ or $AlF_x$. We analyzed both, MTs polymerized using γ-phosphate analogues (*Figure 2A and D*) as well as MTs assembled from GTP and stabilized by these same salts (*Figure 2E and F*), and found no substantial differences (*Table 3*). Our experiments support that $GDP-BeF_3^-$ mimics the GTP-bound state. In addition, MTs stoichiometrically incorporate $Be^7$ (*Carlier et al., 1988*) so, all the subunits within $BeF_3^-$-MTs must be in the $GDP-BeF_3^-$ state. Strikingly, these MTs did not show the 8 nm layer line (*Figure 3A*, line blue), denoting similar intra-dimer and inter-dimer interfaces (*Table 3*), a clear opposition to the expanded-lattice model of the GTP-bound state displayed by the widely accepted GTP-like analogue, GMPCPP. The fitting of the 1 nm layer line revealed very subtle differences at peak maximum when compared to GDP-MTs, which could suggest minor differences on the axial spacing. The equatorial signals revealed that $BeF_3^-$-MTs had a slightly smaller diameter and contained fewer average number of PFs than GDP-MTs (*Table 3*), mainly due to the lack of 14- and 15-PF MTs within the estimated population (*Figure 3C*).

$AlF_x$-MTs showed both, close intra-dimer and inter-dimer distances and roughly similar diameters to $BeF_3^-$-MTs (*Table 3*), suggesting that there is not much of a difference between the GTP- and transition states, either in overall MT dimensions or at the axial interactions. However, $AlF_x$-MTs displayed an additional PF on average (*Table 3*) due to the presence of 14- and 15-PFs MTs.

## Tubulin twist in MTs is a consequence of subtle changes in lattice parameters

Cryo-EM was used to get additional details on GDP-, $BeF_3^-$- and GMPCPP-MTs. Since glycerol at high concentrations is incompatible with cryo-EM and that MT nucleation is difficult in BRB80 buffer in the absence of glycerol, we adopted a seeded assembly strategy. To avoid confusion between GMPCPP- and $BeF_3^-$ lattices, we eluded the use of heterogeneous seeds such as those classically prepared in the presence of GMPCPP. Instead, we first prepared $BeF_3^-$-MT seeds in the presence of 25% glycerol, which were further diluted 1/10 in tubulin without glycerol. MT assembly was found to be efficient under these conditions (*Figure 4—figure supplement 1*), providing suitable conditions for cryo-EM observations. Importantly, depolymerization at 4°C showed a slower rate with respect to GDP-MTs, comparable to that observed with GMPCPP-MTs, further confirming that $BeF_3^-$ stabilizes MTs (even in a non MES-glycerol buffer, *Figure 2E*).

Consistent with the X-ray diffraction analysis, we found that $BeF_3^-$-MTs revealed a majority of 12-PFs MTs (73.1%, *Figure 3C*), GDP-MTs contained essentially a mixed population of 13- and 14-PFs

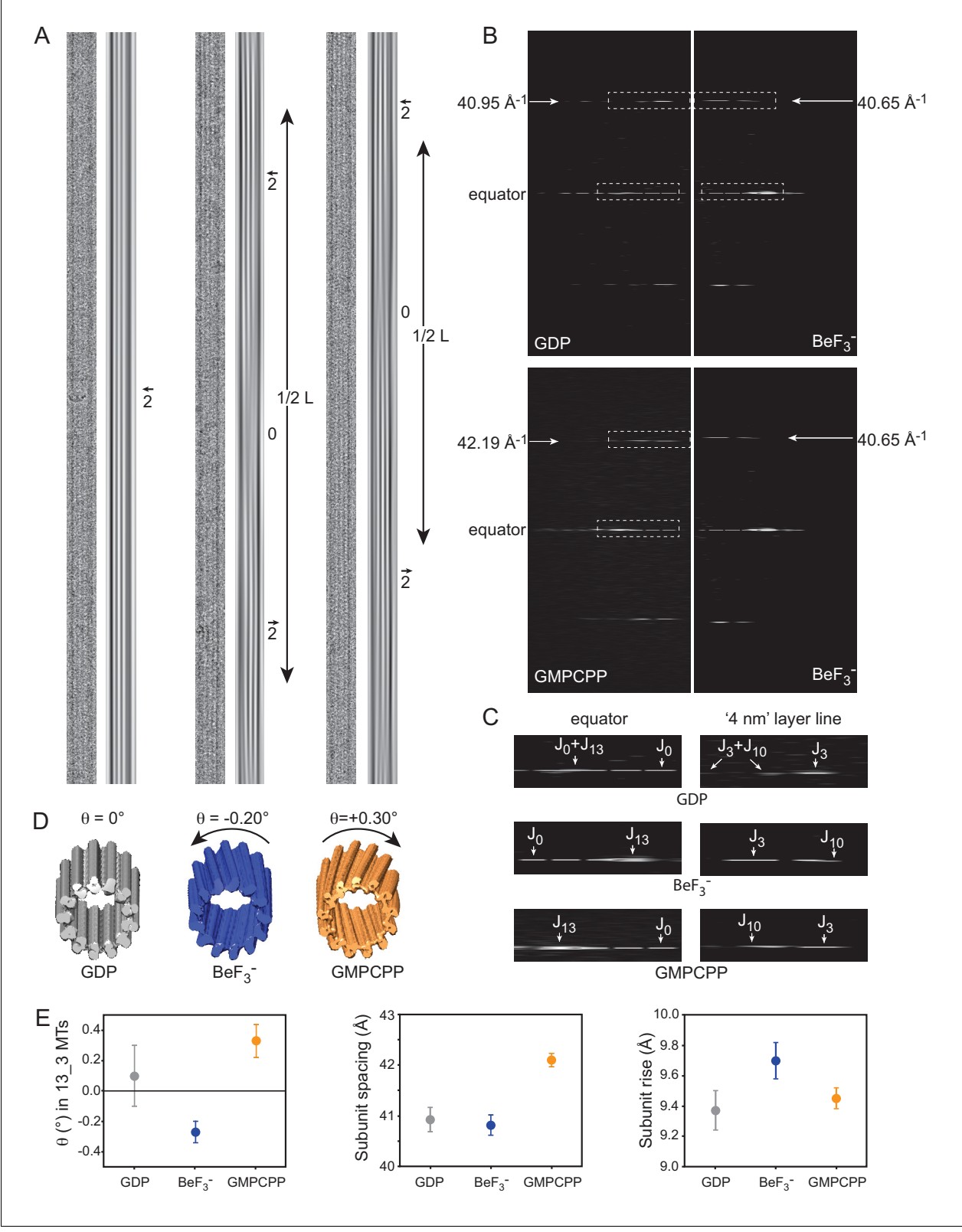

**Figure 4.** Cryo-EM of GDP-, $BeF_3^-$-and GMPCPP-MTs. (**A**) Straightened images of microtubules with 13 PFs (N) and 3-start monomer helices (S), denoted 13_3 (*N_S*) MTs. For each condition: raw image (*left*) and filtered image using the $J_0$ and $J_N$ layer lines in the FFT of the MTs (*right*). Filtered images of 13_3 GDP-MTs display 2 dark inner fringes running parallel to the MT axis and slightly offset from it (arrow), which correspond to PFs from the top and bottom surfaces superposed in projection. In $BeF_3^-$- and GMPCPP-MTs, the 2 fringes make moiré patterns offset from the MT axis on one

*Figure 4 continued on next page*

*Figure 4 continued*

side and the other separated by blurred regions (noted '0' for no internal fringes). The periodicity (L) of these moiré patterns provides a direct measure of their PF skew angle $\theta$ (*Equation 8*). (B) Comparison between the FFTs of the 13_3 GDP- *vs.* the 13_3 BeF$_3^-$-MT (*top*), and the 13_3 GMPCPP- *vs.* the 13_3 BeF$_3^-$-MT (*bottom*) in A. The monomer spacing along PFs ($a$ in *Equation 9*) is given by the position of the $J_3$ layer line in Fourier space, that is 40.95 Å, 40.65 Å, and 42.19 Å for the 13_3 GDP-, BeF$_3^-$- and GMPCPP-MTs in A, respectively. (C) Blow up of the equator and '4 nm layer lines' in 13_3 GDP- (*top*), BeF$_3^-$- (*middle*) and GMPCPP-MTs (*bottom*), corresponding to the boxed regions in B. In 13_3 GDP-MTs, $J_0$ and $J_N$ overlap on the equator, and $J_3$ and $J_{10}$ overlap on the '4 nm layer line' since the PFs are parallel to the MT axis. In 13_3 BeF$_3^-$- and GMPCPP-MTs, $J_{13}$ is away from the equator due to the PF skew. In BeF$_3^-$-MTs, $J_{10}$ is closer to the equator than $J_3$, indicating that the PFs are left-handed (negative skew), while in GMPCPP-MTs, $J_{10}$ is farther apart from the equator than $J_3$, indicating that the PFs are right-handed (positive skew), (*Chrétien et al., 1996*). (D). 3D reconstructions of the 13_3 MTs in A using TubuleJ (*Blestel et al., 2009*). The 3D reconstructions were elongated to the same size as the original images and presented front face at a slight angle with respect to the MT longitudinal axis to emphasize the PF handedness in 13_3 GDP-MT ($\theta = 0°$), BeF$_3^-$-MT ($\theta = - 0.20°$), and GMPCPP-MT ($\theta = + 0.30°$). (E) *Left*: average PF skew angles of 13_3 GDP-MTs ($\theta = + 0.10 \pm 20°$, n = 15), BeF$_3^-$-MTs ($\theta = - 0.27 \pm 0.07°$, n = 9), and GMPCPP-MTs ($\theta = + 0.33 \pm 0.11°$, n = 12). The average PF skew angle of 13_3 GDP_MTs must be lower since a majority of MTs did not show long enough moiré patterns that could be measured at the high magnification used. Therefore, their average PF skew angle is likely closer to 0°. *Middle*: average monomer spacing along PFs in GDP-MTs ($a = 40.93 \pm 0.24$ Å, n = 66), BeF$_3^-$-MTs ($a = 40.82 \pm 0.20$ Å, n = 103), and GMPCPP-MTs ($a = 42.10 \pm 0.13$ Å, n = 25). All $N_S$ microtubule types were included in the analysis. *Right*: inter-PF monomer tubulin rise in GDP-MTs ($r = 9.37 \pm 0.13$ Å, n = 64), BeF$_3^-$-MTs ($r = 9.70 \pm 0.12$ Å, n = 83), and GMPCPP-MTs ($r = 9.45 \pm 0.07$ Å, n = 24) determined using *Equation 9*. MTs with modified lateral interactions (essentially observed in 13_4 and 14_4 MTs in BeF$_3^-$ conditions) were not included in this analysis (*Chrétien and Fuller, 2000*).

The online version of this article includes the following figure supplement(s) for figure 4:

**Figure supplement 1.** additional cryo-EM data.

MTs (49.4% and 45.3%, respectively) and GMPCPP-MTs showed mainly 14-PFs MTs (79.2%), which is consistent with precious observations (*Hyman et al., 1995*). Strikingly, in the presence of BeF$_3^-$, 14-PFs MTs were arranged according to 4-start monomer lattices (14_4), and 13-PFs MTs displayed a mixed population of 13_3 and 13_4 MTs (to be reported elsewhere). While 13_3 GDP-MTs had their PFs essentially parallel to the MT axis (*Figure 4A*, left), both BeF$_3^-$- (*Figure 4A*, middle) and GMPCPP-MTs (*Figure 4A*, right) displayed systematically moiré patterns, implying that their PFs are skewed relative to the MT axis. Determination of the axial spacing of tubulin monomers from the position of the nominal '4 nm layer line' on the Fourier transform (FFT) of straightened MT images (*Figure 4B*) confirmed that GDP- and BeF$_3^-$-MTs displayed compacted lattices (40.95 Å and 40.65 Å for the MTs in *Figure 4A*, respectively), while GMPCPP-MTs displayed an extended lattice (42.13 Å for the 13_3 GMPCCP-MT in *Figure 4A*). Further analysis of the '4 nm layer line' of these MTs (*Figure 4C–D*) and tilting experiments (not shown) revealed that BeF$_3^-$-MTs have left-handed PFs ($J_{10}$ is closer to the equator than J$_3$), while GMPCPP-MTs have right handed PFs ($J_{10}$ is farther apart from the equator than J$_3$ [*Chrétien et al., 1996*]). The average PF skew angles of 13_3 GDP-, BeF$_3^-$-, and GMPCPP-MTs was found to be $+0.10 \pm 0.20°$, $-0.27° \pm 0.17°$, and $+0.33 \pm 0.07°$, respectively (*Figure 4E*). These data can be interpreted in light of the lattice accommodation model (*Chrétien and Fuller, 2000*; *Chrétien and Wade, 1991*) that describes how the MT lattice accommodates changes in PF and/or helical start numbers, as well as modifications of MT lattice parameters. The theoretical PF skew angle ($\theta_{the}$) of any MT type can be calculated according to *Equation 9* (*Table 4*). While 13_3 GDP-MTs are predicted to have a PF skew close to 0°, an increase of the tubulin monomer spacing $a$ from 40.9 Å to 42.1 Å induces a positive skew of $+0.37°$ similar to that found in GMPCPP-MTs. Likewise, a slight modification of the inter-PF subunit rise $r$ (*Equation 10*) from

**Table 3.** Fiber diffraction analysis of MTs in various nucleotide-bound states.

| | GDP | GDP-Tx | GDP-BeF$_3^-$ | GTP-BeF$_3^-$ | GDP- AlF$_x$ | GTP-AlF$_x$ | GMPCPP | GMPPCP | GMPCP |
|---|---|---|---|---|---|---|---|---|---|
| radius (nm) | 11.42 ± 0.10 | 10.87 ± 0.10 | 11.21 ± 0.25 | 11.16 ± 0.10 | 11.25 ± 0.84 | 11.18 ± 0.12 | 11.63 ± 0.10 | 11.62 ± 0.59 | 11.75 ± 0.53 |
| avg. PF number | 12.91 ± 0.10 | 12.37 ± 0.10 | 12.29 ± 0.20 | 12.23 ± 0.10 | 13.43 ± 1.12 | 13.35 ± 0.13 | 13.29 ± 0.08 | 13.03 ± 0.91 | 13.55 ± 0.45 |
| inter-PF distances (nm) | 5.50 ± 0.03 | 5.45 ± 0.01 | 5.67 ± 0.09 | 5.67 ± 0.02 | 5.21 ± 0.46 | 5.22 ± 0.05 | 5.45 ± 0.03 | 5.55 ± 0.38 | 5.40 ± 0.02 |
| avg. monomer length (nm) | 4.06 ± 0.01 | 4.18 ± 0.01 | 4.07 ± 0.01 | 4.07 ± 0.01 | 4.05 ± 0.05 | 4.05 ± 0.01 | 4.18 ± 0.01 | 4.06 ± 0.01 | 4.17 ± 0.01 |
| 1 nm band peak position (nm$^{-1}$) | 6.19 ± 0.01 | 6.02 ± 0.01 | 6.17 ± 0.01 | 6.17 ± 0.01 | 6.20 ± 0.05 | 6.20 ± 0.01 | 6.02 ± 0.01 | 6.20 ± 0.01 | 6.03 ± 0.01 |

[*]Values are Avg ± StdErr.

**Table 4.** Comparison between experimental and theoretical PF skew angles.

| MT type | GDP | | BeF₃⁻ | | GMPCPP | |
|---|---|---|---|---|---|---|
| | 13_3 | 14_3 | 12_3 | 13_3 | 13_3 | 14_3 |
| $\theta_{exp}$ | +0.10 ± 0.21 (n = 15) | −0.62 ± 0.05 (n = 27) | +0.60 ± 0.05 (n = 52) | −0.27 ± 0.07 (n = 9) | +0.33 ± 0.11 (n = 12) | −0.51 ± 0.04 (n = 12) |
| $\theta_{the}$ | +0.05 | −0.74 | +0.61 | −0.31 | +0.37 | −0.44 |

Theoretical PF skew angles ($\theta_{the}$) were calculated according to **Equation 9**, using $a$ = 40.9 Å, $r$ = 9.4 Å, and $\delta x$ = 48.95 Å for GDP-MTs. For GMPCPP MTs, the monomer spacing $a$ was increased to 42.1 Å, and for BeF₃⁻ the inter-PF subunit rise was increased to 9.7 Å.

9.37 Å (GDP-MTs) to 9.70 Å (BeF₃⁻-MTs) is sufficient to account for the negative PF skew (−0.31°) observed in 13_3 BeF₃⁻-MTs. These features were observed in almost all MTs analyzed in this study (see the 12_3 and 14_3 MTs in *Figure 4—figure supplement 1B–E*), although we note that MTs with large changes in inter-PF rise were also observed in the presence of BeF₃⁻ (essentially 13_4 and 14_4 MTs *Chrétien and Fuller, 2000*). This analysis tells us that the positive PF skew observed in GMPCPP-MTs with respect to GDP-MTs is a simple consequence of the increase in tubulin spacing, most likely due to the presence of the methylene group between the α and β phosphates of GMPCPP (see below), and not a conformational change induced by GMPCPP on tubulin.

## MT lattice expansion is not related to its GTP-bound state

GMPCPP has traditionally been considered a good approximation of the GTP-bound state (*Alushin et al., 2014*; *Hyman et al., 1992*; *Kamimura et al., 2016*; *Zhang et al., 2015*), which led to a model in which GTP-tubulin is in an expanded state at the tip of the MTs, and undergoes a compaction followed by rotation between the α and β subunits upon GTP hydrolysis and Pᵢ release, respectively (*Zhang et al., 2015*). Our results suggest that the GTP-state (GDP-BeF₃⁻) is compacted (*Figure 1 and 3*), and that the twist of tubulin in GMPCPP-MTs is a simple consequence of the accommodation of the lattice to the increase in size of GMPCPP-tubulin (*Figure 4*, *Table 4*). We then asked what was the origin of the increase in size in GMPCPP-tubulin. To address this question, we analyzed the structure of MTs assembled in the presence of GMPPCP that is similar to GMPCPP, but with the methylene group sitting between β and γ phosphates instead of α and β phosphates and GMPCP , which is the hydrolyzed version of GMPCPP. X-ray fiber diffraction analysis of GMPCPP-MTs and GMPPCP-MTs (*Table 3*) highlighted that both GTP analogues generated a similar overall cylinder structure showing equivalent PF number composition in which there were no 11-PF MTs and in which 13- and 14-PF MTs prevailed over 12-PF MTs (*Figure 3C*). GMPCPP-MTs displayed a second set of ~8 nm layer lines (*Figure 3A*, *Figure 3—figure supplement 2*), indicating differences between α- and β-tubulin axial spacing, with a monomer repeat that nicely correlates with corresponding measurements from cryo-EM and recently published nude-MT cryo-EM structures (*Zhang et al., 2018*): 4.18 nm vs. 4.22 nm. By contrast, GMPPCP-MTs diffracting images did not show the ~8 nm layer lines (*Figure 3A*, *Figure 3—figure supplement 2*) and hence, a non-expanded lattice and a MT structure equivalent to the BeF₃⁻- and GDP-MTs (*Table 3*). Strikingly, the diffraction pattern of GMPCP-MTs was similar to that of GMPCPP-MTs, showing the 8 nm layer line and an expanded state (*Figure 3*, *Table 3*). Taken all together, these data tell us that: (i) lattice expansion is not related to the presence of γ-phosphate (i.e. a GTP-bound state) and (ii) the methylene in between the α and β phosphates in GMPCPP- and GMPCP-MTs is likely responsible for increasing the size of the tubulin molecule.

We further found that MTs polymerized in the presence of 0.1 mM GTP + 100 µM taxol showed a nucleotide content of GTP(α):GDP(β), 1.08 ± 0.02: 1.04 ± 0.01, similar to other taxane site agents (*Alushin et al., 2014*; *Field et al., 2018*). Since GTP hydrolysis requires a close intra-dimer distance to fulfill the catalytic pocket (*Nogales et al., 1998*; *Oliva et al., 2004*), this result suggests that taxol-induced expansion may happen after GTP hydrolysis. Interestingly, we found that BeF₃⁻- and AlFₓ-MTs assembled in the presence of 100 µM taxol displayed compact lattices (*Figure 3—figure supplement 3*, *Table 5*), bringing up that this drug cannot induce lattice expansion when the interplay between loop T7 and any of these γ-phosphate analogues is retained at the inter-dimer interface.

**Table 5.** Taxol bound BeF$_3^-$- and AlF$_x$-MTs.

| | GDP-BeF$_3^-$ | GTP-BeF$_3^-$ | GDP- AlF$_x$ | GTP-AlF$_x$ |
|---|---|---|---|---|
| Average monomer length (nm) | 4.03 ± 0.01 | 4.03 ± 0.01 | 4.04 ± 0.05 | 4.04 ± 0.01 |
| 1 nm band peak position (nm$^{-1}$) | 6.24 ± 0.01 | 6.24 ± 0.01 | 6.22 ± 0.05 | 6.22 ± 0.01 |

## Discussion

The functions of MTs during cell proliferation and development require continuous rescue and catastrophe events that give rise to motion through dynamic instability. Microtubule-Associated Proteins (MAPs) finely regulate MT function by stabilizing, guiding and destabilizing MT formation, as well as mediating both MT-MT and MT-protein interactions. Plus-end-tracking proteins (+TIPS) are a type of MAP that stabilize or balance MTs; their functions rely on unique structural determinants at the lattice of the MT tip (*Maurer et al., 2011*; *Maurer et al., 2012*; *Zanic et al., 2009*). Thus, understanding these specific structural features is crucial to gain insight into both the mechanisms of MT stabilization and the process of catastrophe.

Incorporation of tubulin dimers into the MT tip forms a cap of not yet hydrolyzed GTP subunits that re-arrange from a naturally bent state to a straight conformation for GTP hydrolysis. The region where this transitions occurs is called the GTP-cap, an uneven structure that can extend along hundreds of nanometers (*Duellberg et al., 2016b*) and ends in an open sheet or flared PFs at the tip (*Chrétien et al., 1995*; *McIntosh et al., 2018*). Recent data point to the formation of one lateral contact as the key interaction favoring length-wise PF growth (*Mickolajczyk et al., 2019*), which occurs 10–50 nm from the growing tip of the PF (~1–7 tubulin heterodimers (*Erickson, 2019*; *McIntosh et al., 2018*). Tubulin straightening is not a single step change but a gradual process (*Figure 5A*), proportional to the increasing formation of lateral interactions in PFs (*Brouhard and Rice, 2018*; *Guesdon et al., 2016*; *Jánosi et al., 1998*). In this process, GTPase activity occurs upon tubulin fully-straighten because this unique conformation allows the interplay of the α-tubulin catalytic loop (T7) and the γ-phosphate at the E-site in β-tubulin (*Nogales et al., 1998*; *Oliva et al., 2004*), meaning the curled tip might be on the GTP-bound state. However, the transitional nature of GTP-cap architecture is difficult to overcome experimentally. Common approaches include the use of homogeneous model states of the MTs. In this work, we used a broad range of nucleotide-bound model MTs to understand their structure throughout the GTPase cycle (*Figure 5A*): active upon assembly GTP-bound (GDP-BeF$_3^-$ and GMPPCP), transitional GDP•P$_i$ (GDP-AlF$_x$), metastable GDP (GTP-assembled MTs), and expanded (GMPCPP, GMPCP and GDP-Tx, see below). Our results merge into a cap model (*Figure 5B*) in which each modeled MT mimics the structural transformations of the filament during elongation, according to the natural biochemical changes caused by GTPase activity.

The existing cap model is based on the commonly accepted GMPCPP analogue as a *bona fide* model of the GTP-bound state and proposes an initial right-handed, expanded lattice (*Alushin et al., 2014*) that rapidly shrinks and changes twist direction upon hydrolysis during the transition state (exemplified by nude GTP-γ-S-MTs (*Zhang et al., 2018*) or doublecortin-bound GDP•P$_i$-MTs *Manka and Moores, 2018*). GMPCPP induces a far stronger assembly than that from GTP (*Hyman et al., 1992*), and MTs are stable because this GTP analogue is not easily hydrolysable, despite of a normal oxygen link between β and γ-phosphates. Whether the enhanced activity is due to the lower GTPase rate or the presence of a methylene link between α and β-phosphates is not clear, though it might be related to the expansion because taxol is also a strong tubulin assembly inducer and produces lattice expansion.

Our approach included γ-phosphate analogues that mimic the chemical structure of phosphates on the activation and transitional states, which we structurally and biochemically analyzed on our tubulin system. These data support that, instead of the commonly accepted model, MTs in the GTP-bound state display a compact configuration (*Figure 5B*, blue). Control experiments with the GTP analogue GMPPCP and with the GDP-analogue GMPCP, confirmed that expansion is due to the presence of the methylene link between α and β-phosphates rather than because of the presence of the γ-phosphate. In our proposed model, the lattice during the transition state of hydrolysis (*Figure 5B*, red) remains unchanged, despite a possible modification at the inter-PF angle. The

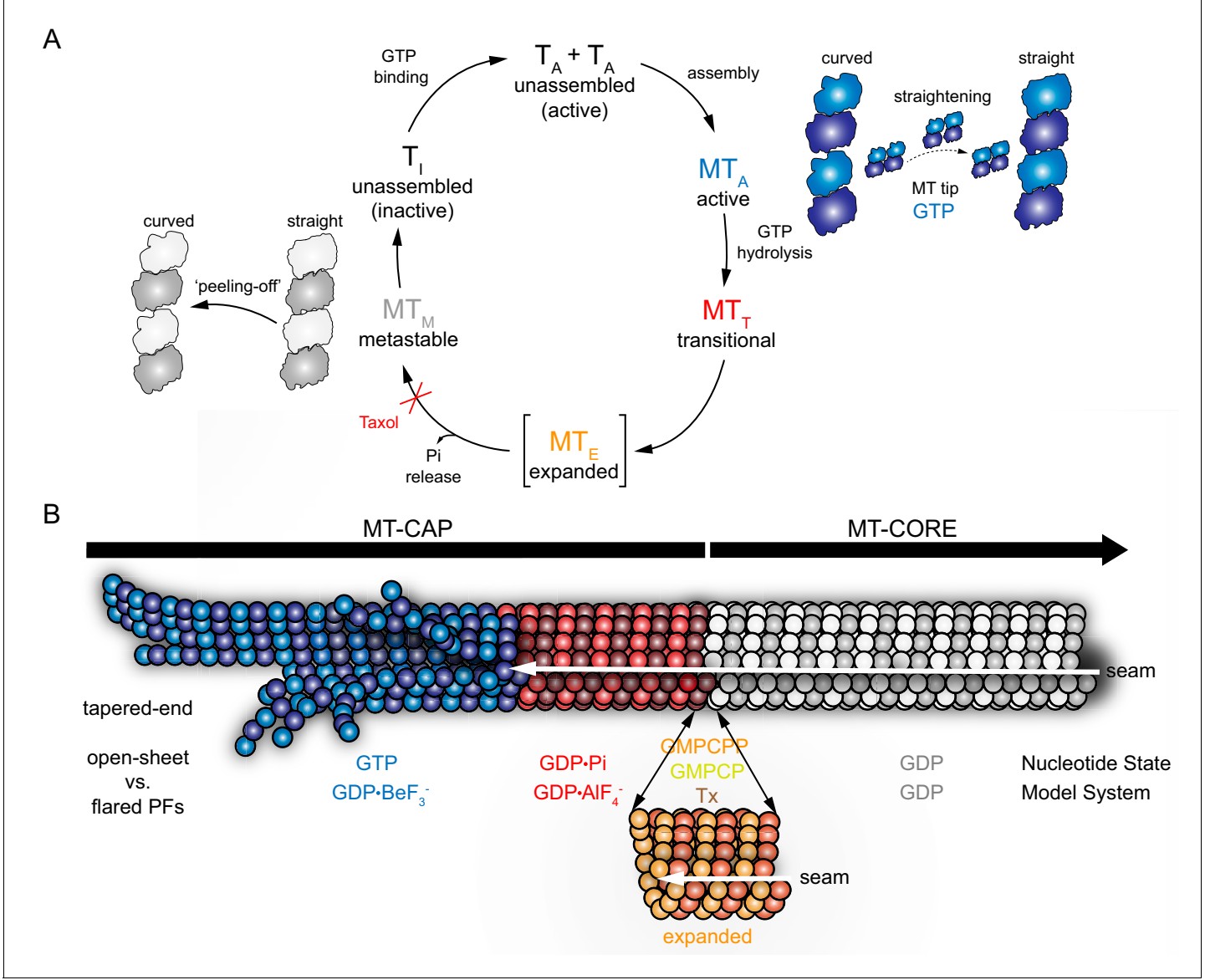

**Figure 5.** Cap model derived from MT model systems. GTP-bound state (BeF$_3^-$, blue), transition state (AlF$_x$, red), expanded state (GMPCPP, GMPCP, GDP-Tx, orange), and GDP-bound state (gray). (**A**) Schematic GTPase related conformational changes within the MT lattice. Tubulin activation upon GTP binding (T$_A$) induces polymerization. During assembly (MT$_A$), the formation of lateral contacts favors tubulin straightening, which allows GTP hydrolysis. GTPase activity drives MT through a transitional state (MT$_T$), where the P$_i$ is at the nucleotide-binding site before it is released. Expansion (orange) may be an intermediate transient step between GDP•P$_i$ and GDP states, which may facilitates P$_i$ release (MT$_E$) and would be blocked in the presence of taxol. GDP-MT (MT$_M$) shrinks through a 'peeling-off' disassembly in which tubulin reverts to the curved conformation, which is inactive (T$_I$) in the GDP-bound state. (**B**) MT model illustrating specific lattice features of the GTPase cycle. This mosaic structure shows that: (i) the GTP-bound tip (blue) contains curved PFs/sheets that come together into a straight lattice due to the formation of lateral contacts, (ii) the post-hydrolysis GDP•P$_i$ lattice (red) retains overall MT structure, (iii) hypothetically, lattice undergoes an energy-consuming expansion phase (orange) that contributes to P$_i$ release, and (iv) in the GDP state (gray) subtle changes on the PF skew distinguish the metastable compact lattice or, (v) lattice reverts into its previous lower energy state (compaction), preventing the structure from returning to the cap architecture.

GDP•P$_i$ state likely sits between the tapered end and the fully closed, cylindrical end. Its average length is difficult to determine but is considered long according to cap-length estimations made using EB proteins (*Duellberg et al., 2016b*; *Seetapun et al., 2012*). Meanwhile, the only difference observed between BeF$_3^-$- and GDP-MTs (*Figure 5B*, gray) is a slight modification of the PF skew (*Figure 4D–E*). Therefore, no substantial lattice changes occur between the cap and the MT core

apart from subtle modifications of the PF skew or the displacement of $Mg^{2+}$ ion coordination across β-γ and α-β phosphates.

It should be mention that the expanded conformation is not limited to mammalian GMPCPP-, GMPCP- and GDP-Tx-MTs. It also has also been captured in GDP-MTs from yeast and worm (*Chaaban et al., 2018*; *Howes et al., 2017*; *von Loeffelholz et al., 2017*), suggesting that this lattice state might not be an artifact. Hence, when or where does this conformational change occur? We propose a plausible explanation considering data arising from the interaction of EB proteins with MTs and recent results on taxanes-binding to the MT cap. EB proteins interact with both, curved and straight regions at MT ends (*Guesdon et al., 2016*), likely recognizing the MT compact lattice, because they bind preferentially to MTs in the presence of GDP-BeF$_3^-$ and GTP-γ-S over GMPCPP (*Maurer et al., 2011*; *Zhang et al., 2015*). Besides, EB proteins form an extended 'comet' that decays exponentially with increasing distance from the tip (*Seetapun et al., 2012*), which highlights their gradual release from the MT wall, with no clue about the structural-chemical determinant inducing such pattern of release. Additionally, it has been shown that fluorescent taxanes bind to the MT region behind the EB 'comet' (*Rai et al., 2019*). From the thermodynamical point of view, if a drug induces a specific conformation, it will bind with higher affinity to this kind of structure, because none of the free energy of binding will be required to induce such conformational change. This implies that taxanes should bind preferentially to expanded lattices because they do induce lattice expansion (*Alushin et al., 2014*). Similarly, EBs might induce lattice compaction (their favored lattice state) in GMPCPP-MTs, causing nucleotide hydrolysis (*Zhang et al., 2018*). Hence, it could be possible that the MT region behind EB-binding site would be expanded, favoring taxanes binding and EBs release. We show that MT lattice is compact among the tubulin GTPase cycle. Therefore, lattice expansion, if existing, might be between two of the main nucleotides bound states (GTP, GDP•P$_i$ and GDP). Since the presence of γ-phosphate analogues (BeF$_3^-$ or AlF$_x$) reverts taxol-induced expansion (*Figure 5A–B*, orange), we speculate that expansion might be an intermediate state between the MT's cap (GDP•P$_i$) and core (GDP) regions, able to facilitate P$_i$ release (*Figure 5B*). Note that, on the stabilization experiments, the γ-phosphate analogues immediately replace the released P$_i$ and so, no expansion of the compact GDP lattice is needed to justify the biochemical stabilization observed.

Considering the narrow region labeled by these taxanes (*Rai et al., 2019*), it could be argued that lattice expansion might involve only few tubulin molecules likely randomly distributed, which has precluded the detection of expansion on cryo-EM images of the MTs tips. If existing, lattice expansion would be likely the point of no return for the GTP-cap as a stabilizing structure. Thereby, once the MT compacts and returns to the GDP-bound state (*Figure 5B*, gray), the remaining energy stored in the lattice would be enough to exert the peeling-off disassembly of PFs, but below the threshold for the MT to revert to the expanded state. Notice that the 'peeling-off' disassembly is highly efficient and uses only 25% of the energy from the GTPase activity (*Driver et al., 2017*) *Brouhard and Rice, 2018*).

The GTP-cap is a unique structure that governs MT dynamics; our model depicts a mosaic architecture that undergoes various maturation steps according to tubulin GTPase activity. While GTP binding activates assembly, GTP hydrolysis produces energy, part of which could be used for P$_i$ release, and limits the length of the cap. The presence of an expanded lattice in GDP-MTs of yeast and worm might be a consequence on the delay or inability to return to the compact state, which could be related to the differences observed on lateral contacts between PFs (*Chaaban et al., 2018*). At the moment, we do not know how is the GTP-MT lattice on these organisms and additional studies are necessary to confirm a compact-to-expanded conformational change upon P$_i$ release, although we cannot rule out that MT lattice could be more expanded in other organisms different than mammals. Beyond this study, high-resolution structural studies will be crucial to analyze the exact nature of protein-protein interactions at the tip of the cap and to fully understand the feedback between structural modifications driven by partner proteins that otherwise are regulating MT dynamics.

## Accession codes

Data deposition: Atomic coordinates and structure factors for the reported crystal structures have been deposited at the Protein Data Bank under the accession code 6gze (BeF$_3^-$) and 6s9e (AlF$_3$).

# Materials and methods

**Key resources table**

| Reagent type (species) or resource | Designation | Source or reference | Identifiers | Additional information |
|---|---|---|---|---|
| Biological Sample | Tubulin alpha | Uniprot | P81947 | purified from calf-brain |
| Biological Sample | Tubulin beta | Uniprot | Q6B856 | purified from calf-brain |
| Gene (*Rattus norvergicus*) | Stathmin-4 | Uniprot | P63043 | Overexpression in E. coli |
| Gene (*Gallus gallus*) | Tubulin-Tyrosine Ligase | Uniprot | E1BQ43 | Overexpression in E. coli |
| Chemical compound, nucleotide | GMPCPP | Jena Bioscience | Jena Bioscience: GpCpp- NU405 | |
| Chemical compound, nucleotide | GMPPCP | Jena Bioscience | Jena Bioscience: GppCp NU-402 | |
| Chemical compound, nucleotide | GMPCP | Jena Bioscience | Jena Bioscience: GpCp NU-414 | |
| Chemical compound, drug | Taxol | Sigma Aldrich | Sigma Aldrich:T7191 | |
| Software, algorithm | XDS | http://xds.mpimf-heidelberg.mpg.de/ | RRID:SCR_015652 | |
| Software, algorithm | AIMLESS | https://www.ccp4.ac.uk/ | RRID:SCR_015747 | |
| Software, algorithm | PHASER | https://www.phenix-online.org/documentation/reference/phaser.html | RRID:SCR_014219 | |
| Software, algorithm | PHENIX | https://www.phenix-online.org/ | RRID:SCR_016736 | |
| Software, algorithm | COOT | https://www.ccp4.ac.uk/ | RRID:SCR_014222 | |
| Software, algorithm | PDBePISA | https://www.ebi.ac.uk/pdbe/pisa/ | RRID:SCR_015749 | |
| software, algorithm | ImageJ | https://imagej.nih.gov/ij/ | RRID:SCR_003070 | |
| Software, algorithm | XRTools | BM26-DUBBLE, ESRF | | |
| Software, algorithm | TubuleJ | https://team.inria.fr/serpico/software/tubulej/ | | |

## GDP-, GMPCP and GMPPCP-tubulin preparation

Purified calf brain tubulin, and chemicals were as described (*Díaz and Andreu, 1993*). Placlitaxel was from Sigma-Aldrich. GMPCP, GMPCPP and GMPPCP were from Jena Biosciences (Jena, Germany). Nucleotides were analyzed by HPLC as described (*Smith and Ma, 2002*). GMPCP was found to be free of GMPCPP and GTP, and GMPPCP was free of GTP. Fully substituted GDP-tubulin was prepared by a two-step interchange procedure; trehalose, $Mg^{+2}$ and free GTP were removed by size exclusion chromatography in a drained centrifuge column of Sephadex G-25 medium (6 × 1 cm), equilibrated in MEDTA buffer (100 mM MES, 1 mM EDTA, pH 6.7) with 1 mM GDP, at 4°C. GDP up to 10 mM was added to the protein, which was incubated for 30 min on ice. Tubulin was freed of

nucleotide excess and equilibrated in MEDTA buffer with 1 mM GDP by a second chromatography in a cold Sephadex G-25 column (15 × 0.9 cm). The nucleotide content of the protein was quantified by protein precipitation and HPLC as described (*Díaz and Andreu, 1993*) and was found to be 0.98 ± 0.07 mol GTP/mol tubulin, and 0.99 ± 0.06 mol GDP/mol tubulin. Alternatively fully substituted GDP-tubulin was obtained from GTP-tubulin by hydrolysis. GDP tubulin was equilibrated in a 20 × 1 cm Sephadex G-25 column equilibrated in MEDTA buffer with 3.4 M Glycerol. The nucleotide content of the protein was measured as above and was found to be 0.24 ± 0.02 mol GDP/mol tubulin and 1.75 ± 0.06 mol GTP/mol tubulin. The protein was diluted to 30 µM and supplemented with 6 mM $Mg^{+2}$ (final pH 6.5) to induce polymerization into MTs by incubating at 37°C. The assembly was monitored turbidimetrically at 350 nm. Protein assembled into MTs (as checked by electron microscopy) and rapidly disassembled after hydrolysis of the bound nucleotide. Samples were taken at the peak of assembly (Sample A) and after complete disassembly (Sample B). The nucleotide content of the protein was measured as above and was found to be 1.01 ± 0.02 mol GDP/mol tubulin and 0.99 ± 0.03 mol GTP/mol tubulin (Sample A) and 0.96 ± 0.04 mol GDP/mol tubulin and 1.00 ± 0.03 mol GTP/mol tubulin (Sample B). Prior to assembly, tubulin samples were centrifuged at 100,000 *g*, 4°C, for 10 min using TL100.2 or TL100.4 rotors in a Beckman Optima TLX centrifuge to remove aggregates. GMPCP and GMPPCP tubulin preparation were performed by a two-step protocol based on *Mejillano et al. (1990)*. First, 20 mg of lyophilized calf brain tubulin were resuspended in PM buffer (80 mM K-PIPES, 1 mM EGTA, 1 mM dithiothreitol, 0.2 mM Tris, pH 6.8) containing 2 mM GTP and 1.5 mM $MgCl_2$, in order to get the GDP bound to the protein substituted by GTP due to its higher affinity in the presence of $Mg^{+2}$ ions (*Correia et al., 1987*). Then, the free $Mg^{+2}$ and GTP were removed by chromatography in a drained column of Sephadex G-25 (GE Healthcare) (6 × 1 cm), equilibrated in PM buffer containing 50 nM of the desired analogue, two times washing steps by filtration using Amicon MWCO 50 (Merck-Millipore), and concentrated to 700–800 µl. The protein was then passed through 0.45 µm cellulose acetate microfuge column filters (Costar) to remove aggregates. Then 5 mM of the desired nucleotide analogue was added to displace the GTP bound (due to its lower affinity in the absence of $Mg^{+2}$) and the protein was incubated for 30 min at 25°C. The protein was washed and concentrated as before to 600–700 µl and filtered again on a 0.45 µm cellulose acetate microfuge column filter to a final concentration 100–150 µM. After filtration, the protein was HPLC analyzed as described (*Smith and Ma, 2002*) and was found to be 90% loaded with the desired nucleotide at the E-site. In order to induce MT assembly, tubulin was supplemented with 2 mM GMPCPP, GMPCP or GMPPCP and 3 mM $MgCl_2$ prior to polymerization.

## Crystallization and crystal structure determination

The stathmin-like domain of RB3 and the chicken TTL protein preparations were done as described previously (*Prota et al., 2013b*; *Ravelli et al., 2004*). For the $T_2$R-TTL complex, tubulin (8 mg/mL), TTL (17 mg/mL) and RB3 (26 mg/mL) were mixed and concentrated (Amicon MWCO 10) at 4°C to a final complex concentration of 20 mg/ml. The concentrated mixture was supplemented with 10 mM DTT, 0.1 mM GDP, 1 mM AMPCPP and 5 mM $BeF_3^-$ (in $T_2$RT- $BeF_3^-$ complex), or 1.5 mM $AlCl_3$ and 2 mM $HKF_2$, or 500 µM $AlCl_3$, 2 mM $HKF_2$ (in $T_2$RT- $AlF_x$ complex), before setting crystallization experiments. Initial crystallization conditions were determined from previous structures (*Prota et al., 2014*; *Prota et al., 2013a*; *Prota et al., 2016*) using the sitting-drop vapor diffusion technique with a reservoir volume of 200 µl and a drop volume of 1 µl of complex and 1 µl of reservoir solution at 20°C. Crystal-producing conditions were further optimized using the hanging drop vapor diffusion method with a reservoir volume of 500 µl and a drop volume of 1 µl of complex and 1 µl of reservoir solution. Native $T_2$R-TTL-$BeF_3^-$ complex was crystallized in 0.1 M MES/0.1 M Imidazole pH 6.5, 0.03 M $CaCl_2$/0.03 M $MgCl_2$, 5 mM L-tyrosine, 8.8% glycerol, 5.5% PEG4000. $T_2$R-TTL-$AlF_x$ complex was crystallized in 0.1 M MES/0.1 M Imidazole pH 6.5, 0.03 M $CaCl_2$/0.03 M $MgCl_2$, 5 mM L-tyrosine, 5% glycerol, 5.5% PEG4000 500 µM $AlCl_3$, 2 mM $HKF_2$ (alternatively 1.5 mM $AlCl_3$ and 2 mM $HKF_2$ were used). Plates were kept at 20°C and crystals appeared within the next 24 hr. Prior flash-cooling in liquid nitrogen, crystals were cryo-protected using 10% PEG4000, increasing glycerol concentrations (16% and 20%), and were supplemented with 7.7 mM $BeF_3^-$ or 500 µM $AlCl_3$ and 2 mM $HKF_2$ or 1.5 mM $AlCl_3$ and 2 mM $HKF_2$ depending on the growing conditions. X-ray diffraction data were collected on beamline ID23-1 of the European Synchrotron Radiation Facility (ESRF) and beamline XALOC of ALBA Synchrotron. Diffraction intensities were indexed and integrated using XDS

(*Kabsch, 2010*), and scaled using AIMLESS (*Winn et al., 2011*). Molecular replacement was performed with PHASER (*McCoy et al., 2007*) using the previously determined structure (PDB 4o2b) as a search model. Structures were completed with cycles of manual building in COOT (*Emsley et al., 2010*) and refined in PHENIX (*Adams et al., 2010*), which allowed the determination of alternative conformations on different residues at main chains. Data collection and refinement statistics are summarized in *Table 1*. The ligand interfaces of this structure and other tubulin structures in the curved (PDBs 5xp3, 3ryh, 4i55) or straight (PDBs 3jat, 3jal, 3jar, 3jak, 6ecx) conformations were analyzed using PDBe PISA (*Krissinel and Henrick, 2007*). Omit maps were calculated in PHENIX using Cartesian annealing and harmonic retrains on the omitted atoms ($BeF_3^-$ or $AlF_3$).

## Microtubule assembly and Cr determination

Tubulin (calf brain purified or fully GDP- exchanged as described above) was exchanged into MEDTA buffer and mixed at a 60:40 ratio with a 8.5 M glycerol, 100 mM MES, 1 mM EDTA buffer pH 6.7 (final buffer concentrations 100 mM MES, 3.4 M glycerol, 1 mM EDTA, 0.6 mM GDP, pH 6.7). These samples were supplemented with $MgCl_2$ and increasing concentrations of $BeF_3^-$, 3 mM $HKF_2$ plus increasing concentrations $AlCl_3$, 30 µM GTP plus increasing concentrations of $BeF_3^-$ or 30 µM GTP plus 3 mM $HKF_2$ plus increasing concentrations of $AlCl_3$. The solutions were warmed up to 37°C, and incubated as long as needed to reach polymerization equilibrium. The assembly was monitored by turbidity at a wavelength of 350 nm. Alternatively, the polymers formed were sedimented at 100,000 *g* for 20 min in a TLA 100 rotor equilibrated at 37°C. Supernatants were separated by aspiration, and pellets were resuspended in a 1% SDS, 10 mM phosphate buffer. Tubulin concentration in pellets and supernatants was measured spectrofluorometrically by excitation at 280 nm and emission at 320 nm (*Buey et al., 2005*) employing a Shimadzu RF-540 fluorometer (excitation and emission slits, 5 nm) calibrated with standards of known concentration prepared from the same tubulin. Apparent polymer growth equilibrium constants were estimated as the reciprocal of the critical concentrations for polymerization determined at several total protein concentrations (*Oosawa, 1975*).

## MT shear-flow alignment and X-ray fiber diffraction experiments

X-ray fiber diffraction data were collected on beamlines BL11-NDC-SWEET and BL40XU of ALBA (Spain) and SPring-8 (Japan) synchrotrons. Purified bovine brain tubulin (5 mg) was diluted in 500 µL to a final concentration of 100 µM on PM buffer containing 3 mM $MgCl_2$ and either 2 mM GTP, 0.5 mM GMPCPP, 5 mM GMPCP, 5 mM GMPPCP, 2 mM GTP + 10 mM $BeF_3^-$, 2 mM GTP + 500 µM $AlCl_3$ + 2 mM $HKF_2$, 2 mM GTP + 200 µM taxol, 2 mM GTP + 10 mM $BeF_3^-$ + 200 µM taxol or 2 mM GTP + 500 µM $AlCl_3$ + 2 mM $HKF_2$ + 200 µM taxol. GDP-tubulin was obtained from GTP-tubulin by hydrolysis as described above, and then supplemented with 1 mM GDP + 10 mM $BeF_3^-$ or 1 mM GDP + 500 µM $AlCl_3$ + 2 mM $HKF_2$. All samples were incubated for 20 min at 37°C to induce the maximum fraction of polymerized tubulin, and then were mixed in a 1:1 vol ratio with PM buffer at 3 mM $MgCl_2$ containing 2% methylcellulose (MO512; Sigma-Aldrich). Final concentration of nucleotides, salts and drugs were 1 mM GTP, 0.25 mM GMPCPP, 2.5 mM GMPCP/GMPPCP, 1 mM GTP + 10 mM $BeF_3^-$, 1 mM GTP + 500 µM $AlCl_3$ + 2 mM $HKF_2$ (salts were not diluted because methylcellulose was further supplemented) and 100 µM taxol. Samples were centrifuged 10 *s* at 2000 *g* to eliminate air bubbles and were transferred to the space between a mica disc and a copper plate with a diamond window in the shear-flow device (*Figure 3—figure supplement 1*, (*Kamimura et al., 2016*; *Sugiyama et al., 2009*). The device was kept at 37°C during measurements and the mica disc spun at 10 rps to achieve the shear flow alignment. Several experiments were recorded using different sample-to-detector distances and wavelengths, collecting 150 *s* exposure time images on a PILA-TUS3S-1M detector. Background images were acquired in the same conditions, using PM buffer containing 3 mM $MgCl_2$ with 1% methylcellulose without tubulin. A total of 16–24 diffraction images were averaged for each condition and background subtracted using ImageJ (version 1.51j8; Wayne Rasband, National Insitutes of Health, Bethesda, USA). Angular image integrations were performed using the XRTools software (obtained upon request from beamline BM26-DUBBLE of the ESRF), where the spatial calibration was obtained from Ag-Behenate powder diffraction. For data analysis, we considered MTs as cylinders with a three-start helical pattern. The diffraction pattern of MTs comprises layer lines (*l*) each defined by a group of Bessel functions of order *n*. Their structural factor F in the reciprocal space (R) is described by *Equation 1* (*Klug et al., 1958*):

$$F_{l,n}(R) = J_n(2\pi r_m R)f(R) \tag{1}$$

where $J_n$ is the $n^{th}$ Bessel function, $r_m$ is the radius of a MT with $m$ PFs and $f(R)$ is the structural factor of the sphere defined by *Equation 2* (*Malinchik et al., 1997*):

$$f(R) = 4\pi r_t^3 \frac{\sin(2\pi r_m R)\cos(2\pi r_m R)}{(2\pi r_m R)^3} \tag{2}$$

This expression is used to include the structural factor of the tubulin wall in the calculation where $r_t$ is the radius of the tubulin monomer considered as a sphere, with a value of 2.48 nm (*Kamimura et al., 2016*). For radial structural parameters (average MT radius, $r_m$; average PF number, $m_a$; and average PF distances, $d_m$), we analyzed the central-equatorial intensity profile ($l = 0$) (*Figure 3—figure supplement 1* blue line). The relationship between diffraction intensity and the structural factor is represented in the expression *Equation 3* (*Amos and Klug, 1974*; *Cochran et al., 1952*):

$$<I_{l,n}(R)> = \left|F_{l,n}(R)\right|^2 \tag{3}$$

The intensity of this layer line $I(R)$ at a reciprocal distance ($R$) results from a MT mixed population with different PF numbers ($m$), often from 10 to 15. Considering $w_m$ as the fraction of MTs with $m$ PFs, the resulting intensity was deconvolved as the sum of intensities of individual structure functions $F_{l,n}$ (*Equation 4*)

$$_{l,n}(R)> = \sum_m w_m (\sum_n \left|F_{o,n}(R)\right|^2) \tag{4}$$

In the $l = 0$ layer line, diffraction intensity is explained by $J_0$ and $J_n$ where $n = m$ (*Oosawa, 1975*). Therefore, from *Equation 4* we obtain the following expression (*Equation 5*):

$$<I_{l,m}(R)> = \sum_m w_m (\left|F_{0,0}(R)\right|^2 + \left|F_{0,m}(R)\right|^2) \tag{5}$$

From *Equation 1* and *Equation 5* we derive the *Equation 6*:

$$<I_{l,m}(R)> = \sum_m w_m \left((f(R)J_0(2\pi r_m R))^2 + (f(R)J_m(2\pi r_m R))^2\right) \tag{6}$$

This equation was used for iterative fitting by least squares of the experimental intensities using the Solver function in Excel (Microsoft, 2010 version). For inter-PF distance determination ($d_m$), $r_m$ was used as the apothem of an $m$-apex MT in which the linear distance between them is calculated according to *Equation 7*:

$$d_m(r) = 2r_m \sin\left(\frac{\pi}{m_a}\right) \tag{7}$$

The standard error of these calculated values was determined from the ratio between the experimental maximum intensities values of $J_0$ and $J_n$ and their standard deviation. For the determination of the average monomer lengths, we analyzed the intensity profile of the central-meridional signals (*Figure 3—figure supplement 1*, red line). The 4$^{th}$ harmonic of the first layer-line ($l = 4$)) was fitted to a single-peaked Lorentzian function using Sigma-Plot software (version 12.0), in which the position of the maximum of intensity corresponds to ¼ of the average monomer length at the reciprocal space. The standard error was obtained from the standard deviation of the regression applied.

## Negative-staining and cryo-electron microscopy

For negative-stain experiments, samples of the assembled tubulin were routinely adsorbed to carbon-coated Formvar films on 300-mesh copper grids, stained in 2% uranyl acetate and observed with a JEOL JEM-1230 at 40,000 K magnification with a digital camera CMOS TVIPS TemCam-F416.

For cryo-EM experiments, tubulin was isolated from porcine brain by two cycles of assembly disassembly (*Castoldi and Popov, 2003*), followed by a final cycle in the absence of free GTP (*Mitchison and Kirschner, 1984b*). GDP-tubulin was obtained in BRB80 and stored at $-80°C$ before

use. Spectrophotometry was used to determine suitable conditions for cryo-EM. GDP-MTs were polymerized at a final tubulin concentration of 40 µM in BRB80 (80 mM K-Pipes, 1 mM EGTA, 1 mM $Mg^{2+}$, pH 6.8 with KOH), 1 mM GTP, 35°C. GMPCPP-MTs were polymerized at 10 µM tubulin concentration in BRB80, 0.1 mM GMPCPP, 35°C. GDP/$BeF_3^-$-MTs were assembled using a seeded strategy. GDP/$BeF_3^-$-MT seeds were polymerized at 40 µM tubulin concentration in BRB80, 25% glycerol, 20 mM NaF, 5 mM $BeSO4^-$, 35°C for 1 hr. MTs were sheared by sonication (30 *s*) followed by up-and-down pipetting, and were diluted 1/10 in pre-warmed tubulin at 40 µM with the same buffer composition but in the absence of glycerol, giving a final glycerol concentration of 2.5% suitable for cryo-EM experiments. The presence of aggregates was checked by switching the temperature to 4°C once the MTs reached steady state. Cryo-EM grids were typically prepared after ~1 hr of assembly. Tobacco Mosaic Virus (TMV) was added to the suspensions before polymerization to be used as internal calibration standards. Four µl samples were pipetted and deposited at the surface of holey-carbon coated grids (R3.5/1, Quantifoil) in an automatic plunge freezer (EM-GP, Leica) under temperature and humidity controlled conditions. Grids were stored in $LN_2$ before use. Specimen grids were loaded onto a cryo-holder (model 626, Gatan), and were observed in a 200 kV electron microscope (Tecnai G2 Sphera, FEI) equipped with a $LaB_6$ cathode and a 4k × 4 k CCD camera (USC4000, Gatan). Images were taken at a nominal magnification of 50,000 X using a −1.5 µm to −3 µm defocus range. Images were further calibrated using the 2.3 nm layer line of the TMVs added to the suspensions, which gave a pixel size of 2.16 ± 0.01 Å (n = 27). Individual MT images were straightened using TubuleJ (*Blestel et al., 2009*), which allows a semi-automatic determination of MT centers using the phase information on $J_0$ in the FFT of the MT images. The PF skew angle '$\theta_{exp}$' was determined from the length of the moiré patterns (*L*, or ½ *L* in the case of 13_3 MTs) in the $J_0+J_N$ filtered images of the MTs using:

$$\theta exp = sin^{-1}\left(\frac{\delta x}{L}\right) \qquad (8)$$

where $\delta x$ denotes the lateral separation between PFs ($\delta x$ = 48.95 Å). This latter value was estimated from the increase in MT diameter with PF number in 3D maps of MTs (*Sui and Downing, 2010*). The sign of $\theta_{exp}$ was deduced from the analysis of the respective positions of $J_S$ and $J_{N-S}$ on the '4 nm layer lines' on the FFT of the MT images. Series of tilted images were also used (*Chrétien et al., 1996*) in cases where the separation between $J_S$ and $J_{N-S}$ was minimal (e.g. 13_3 MTs). The monomer spacing along PFs *a* was determined from the position of the $J_S$ layer line in the FFT of the MT images (*S*: monomer helical rise). The theoretical PF skew angle $\theta_{the}$ on any N_S MT can be calculated using the following formula (*Chrétien and Fuller, 2000*):

$$\theta the = \tan^{-1}\left(\frac{1}{\delta x} - \left(\frac{Sa}{N} - r\right)\right) \qquad (9)$$

The inter-PF rise *r* was calculated according to the following formula (*Chrétien and Fuller, 2000*):

$$r = \frac{Sa}{N} - \frac{\delta x^2}{L\sqrt{1 - \left(\frac{\delta x}{L}\right)^2}} \qquad (10)$$

where *N* denotes the PF number of MTs.

## Microtubule nucleotide content

Fully GDP-exchanged tubulin was obtained as described before in MEDTA with 3.4 M Glycerol, 0.1 mM GDP, 6 mM $MgCl_2$, pH 6.7. 40 µM GDP-tubulin in the presence of 0.1 mM GTP or 0.1 mM GTP and 50 µm Taxol was incubated at 37°C for 30 min. MTs were sedimented as above and pellets were resuspended in 10 mM phosphate buffer. Nucleotide extraction and HPLC separation were then carried out to determine the concentration of each nucleotide as described (*Díaz and Andreu, 1993*).

## Acknowledgements

We thank Ganadería Fernando Díaz for calf brains supply and staff of beamlines ID23-1 and BM26 (ESRF, Grenoble, France), BL11-NCD-SWEET and XALOC (ALBA, Cerdanyola del Vallès, Spain), and

BL40XU (SPring-8, Japan) for their support. This work was supported by Ministerio de Economia y Competitividad grants BFU2013-47014-P to MAO and BFU2016-75319-R to FDP (both AEI/FEDER, UE); Ministerio de Ciencia e Innovación RYC-2011–07900 to MAO; European Union H2020-MSCA-ITN-ETN/0582 ITN TubInTrain to FDP and AEP; Swiss National Science Foundation (31003A_166608) to MOS; JSPS KAKENHI (16K07328/17H03668) to SK; (FSE) ANR-16-CE11-0017-01 to DC and LD. JE-G was supported by Ministerio de Educación, Cultura y Deporte FPU15-03140 and SK by French Ministry of Higher Education of Research and Innovation (IGDR). Cryo-EM data were adquired on the MRic platform (Univ. Rennes, CNRS, Inserm, BIOSIT-UMS 3480, US_S 018, F-35000 Rennes, France). The authors acknowledge networking contribution by the COST Action CM1407 "Challenging organic syntheses inspired by nature - from natural products chemistry to drug discovery".

## Additional information

### Funding

| Funder | Grant reference number | Author |
| --- | --- | --- |
| Ministerio de Economía y Competitividad | BFU2013-47014P | Maria A Oliva |
| Ministerio de Economía y Competitividad | RYC-2011-07900 | Maria A Oliva |
| Ministerio de Economía y Competitividad | BFU2016-75319-R | J Fernando Díaz |
| H2020 European Research Council | H2020-MSCA-ITN-EJD-860070 | Andrea E Prota<br>J Fernando Díaz |
| Swiss National Science Foundation | 31003A_166608 | Michel O Steinmetz |
| Japan Society for the Promotion of Science | KAKENHI 16K07328/17H03668 | Shinji Kamimura |
| Agence Nationale de la Recherche | ANR-16-C11-0017-01 | Denis Chrétien |
| SPring-8 Proposal | 2016B1182 | Shinji Kamimura |
| SPring-8 Proposal | 2019B1365 | Shinji Kamimura |

The funders had no role in study design, data collection and interpretation, or the decision to submit the work for publication.

### Author contributions

Juan Estévez-Gallego, Formal analysis, Investigation, Methodology; Fernando Josa-Prado, Isabel Barasoain, Validation, Investigation; Siou Ku, Ruben M Buey, Francisco A Balaguer, Daniel Lucena-Agell, Toshiki Yagi, Hiroyuki Iwamoto, Formal analysis, Investigation; Andrea E Prota, Funding acquisition, Investigation; Christina Kamma-Lorger, Resources, Investigation; Laurence Duchesne, Conceptualization, Supervision, Funding acquisition; Michel O Steinmetz, Resources, Funding acquisition, Project administration; Denis Chrétien, Conceptualization, Formal analysis, Supervision, Funding acquisition, Writing - review and editing; Shinji Kamimura, Funding acquisition, Investigation, Methodology; J Fernando Díaz, Conceptualization, Formal analysis, Supervision, Funding acquisition, Investigation, Project administration, Writing - review and editing; Maria A Oliva, Conceptualization, Data curation, Formal analysis, Supervision, Funding acquisition, Validation, Investigation, Methodology, Writing - original draft, Project administration, Writing - review and editing

### Author ORCIDs

Juan Estévez-Gallego http://orcid.org/0000-0003-3889-8488
Fernando Josa-Prado http://orcid.org/0000-0002-6162-3231
Ruben M Buey http://orcid.org/0000-0003-1263-0221
Denis Chrétien https://orcid.org/0000-0001-8261-4396

J Fernando Díaz (ID) https://orcid.org/0000-0003-2743-3319
Maria A Oliva (ID) https://orcid.org/0000-0002-2215-4639

Decision letter and Author response
Decision letter https://doi.org/10.7554/eLife.50155.sa1
Author response https://doi.org/10.7554/eLife.50155.sa2

## Additional files

### Supplementary files

• Transparent reporting form

### Data availability

Diffraction data have been deposited in PDB under the accession codes 6gze and 6s9e.

The following datasets were generated:

| Author(s) | Year | Dataset title | Dataset URL | Database and Identifier |
|---|---|---|---|---|
| Oliva MA, Diaz JF | 2020 | Tubulin-GDP.BeF complex | http://www.rcsb.org/structure/6gze | RCSB Protein Data Bank, 6gze |
| Oliva MA, Estevez-Gallego J, Diaz JF, Prota AE, Steinmetz MO, Balaguer FA, Lucena-Agell D | 2020 | Tubulin-GDP.AlF complex | http://www.rcsb.org/structure/6s9e | RCSB Protein Data Bank, 6s9e |

The following previously published datasets were used:

| Author(s) | Year | Dataset title | Dataset URL | Database and Identifier |
|---|---|---|---|---|
| Prota AE, Bargsten K, Zurwerra D, Field JJ, Diaz JF, Altmann KH, Steinmetz MO | 2012 | Crystal structure of tubulin-stathmin-TTL complex | https://www.rcsb.org/structure/4i55 | RCSB Protein Data Bank, 4i55 |
| Nawrotek A, Knossow M, Gigant B | 2011 | GMPCPP-Tubulin: RB3 Stathmin-like domain complex | https://www.rcsb.org/structure/3ryh | RCSB Protein Data Bank, 3ryh |
| Wang Y, Yang J, Wang T, Chen L | 2017 | Crystal structure of apo T2R-TTL | https://www.rcsb.org/structure/5xp3 | RCSB Protein Data Bank, 5xp3 |
| Zhang R, Nogales E | 2015 | Cryo-EM structure of GMPCPP-microtubule (14 protofilaments) decorated with kinesin | https://www.rcsb.org/structure/3jat | RCSB Protein Data Bank, 3jat |
| Zhang R, Nogales E | 2015 | Cryo-EM structure of GMPCPP-microtubule co-polymerized with EB3 | https://www.rcsb.org/structure/3jal | RCSB Protein Data Bank, 3jal |
| Zhang R, Nogales E | 2015 | Cryo-EM structure of GDP-microtubule co-polymerized with EB3 | https://www.rcsb.org/structure/3jar | RCSB Protein Data Bank, 3jar |
| Zhang R, Nogales E | 2015 | Cryo-EM structure of GTPgammaS-microtubule co-polymerized with EB3 (merged dataset with and without kinesin bound) | https://www.rcsb.org/structure/3jak | RCSB Protein Data Bank, 3jak |
| Manka SW | 2017 | Cryo-EM structure of GDP.Pi-microtubule rapidly co-polymerised with doublecortin | https://www.rcsb.org/structure/6evx | RCSB Protein Data Bank, 6evx |
| Nassar N, Hoffman G, Clardy J, Cerione R | 1998 | TRANSITION STATE COMPLEX FOR GTP HYDROLYSIS BY CDC42: COMPARISONS OF THE HIGH RESOLUTION STRUCTURES FOR CDC42 BOUND TO THE ACTIVE AND CATALYTICALLY | https://www.rcsb.org/structure/2ngr | RCSB Protein Data Bank, 2ngr |

| | | | | |
|---|---|---|---|---|
| | | COMPROMISED FORMS OF THE CDC42-GAP. | | |
| Nassar N, Hoffman GR, Clardy JC, Cerione RA | 1998 | CRYSTAL STRUCTURE OF THE CDC42/CDC42GAP/ALF3 COMPLEX | https://www.rcsb.org/structure/1grn | RCSB Protein Data Bank, 1grn |
| Ghosh A, Praefcke GJK, Renault L, Wittinghofer A, Herrmann C | 2005 | Crystal-structure of the N-terminal Large GTPase Domain of human Guanylate Binding protein 1 (hGBP1) in complex with GDP/AlF3 | https://www.rcsb.org/structure/2b92 | RCSB Protein Data Bank, 2b92 |
| Pan X, Eathiraj S, Munson M, Lambright DG | 2007 | Crystal Structure of Gyp1 TBC domain in complex with Rab33 GTPase bound to GDP and AlF3 | https://www.rcsb.org/structure/2g77 | RCSB Protein Data Bank, 2g77 |
| Yu Q, Yao Q, Wang D-C, Shao F | 2013 | Crystal structure of LepB GAP domain from Legionella drancourtii in complex with Rab1-GDP and AlF3 | https://www.rcsb.org/structure/4jvs | RCSB Protein Data Bank, 4jvs |
| Mishra AK, Delcampo CM, Collins RE, Roy CR, Lambright DG | 2013 | Crystal Structure of lepB GAP core in a transition state mimetic complex with Rab1A and ALF3 | https://www.rcsb.org/structure/4iru | RCSB Protein Data Bank, 4iru |

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
