## [Decision Letter]

**Acceptance summary:**

The manuscript by Estévez-Gallego et al. sets out to investigate the conformational state of tubulin in different nucleotide states, aiming to shed light on the mechanism of dynamic instability in microtubules. They present crystal structures of curved tubulin dimers in the presence of two fluoride salts that act as γ-phosphate mimics when combined with GDP. Based on the conformation of the active site, they conclude that GDP-BeF_3_-accurately mimics the GTP-bound state of tubulin, and GDP-AlF_3_-mimics a transitional GDP-Pi state. They show that the two fluoride salts both facilitate microtubule polymerization and stabilization, similar to the commonly used γ-phosphate mimic GMPCPP. They then use X-ray fiber diffraction and cryoEM imaging to measure the lattice spacing of microtubules assembled with different nucleotides. Based on these measurements and the snapshots in the crystal structures, they present a model for microtubule polymerization.

Their key finding is that the lattice spacing in both the GTP-mimic state (GDP-BeF_3_-) and GDP-Pi-mimic state (GDP-AlF_3_-) is equivalent to that in the GDP state (~8.1nm/dimer). This contradicts the prevalent model in the field, in which the GTP-bound lattice has an extended spacing of ~8.3nm/dimer, and collapses to ~8.1nm/dimer upon hydrolysis and/or phosphate release. These conformational changes are thought to weaken lateral interactions, leading to depolymerisation being favoured in the GDP-bound state. As such, the results presented in this study challenge this model, and are such of great interest to the field.

**Decision letter after peer review:**

[Editors’ note: the authors submitted for reconsideration following the decision after peer review. What follows is the decision letter after the first round of review.]

Thank you for submitting your work entitled "Structural model for differential cap maturation at growing microtubule ends" for consideration by *eLife*. I regret to inform you that your work will not be considered further for publication in *eLife* at this time. While there was overall enthusiasm for the work, the reviewers felt that additional work is necessary to support the conclusions and that the work needed would take more than two months to complete. Since it is the policy at *eLife* not to request additional experiments that would take more than two months, we are rejecting the manuscript at this time. Given the favorable opinion of the work overall, we would be happy to consider another submission that includes the additional work suggested.

Your article was assessed by three peer reviewers, one of whom is a member of our Board of Reviewing Editors, and the evaluation has been overseen by a Senior Editor. The following individual involved in review of your submission has agreed to reveal their identity: Andreas Hoenger (Reviewer #3).

As you will see from the comments below, the reviewers found much to like about your manuscript, but overall have a number of concerns. At the moment there are too many points that need to be addressed for the time frame that *eLife* allows for revisions. From the reviewers’ discussion, the question of cryoEM analysis came up. All reviewers felt that this would add strong support to your model.

Reviewer #1:

The manuscript by Estévez-Gallego et al. sets out to investigate the conformational state of tubulin in different nucleotide states, aiming to shed light on the mechanism of dynamic instability in microtubules. They present crystal structures of curved tubulin dimers in the presence of two fluoride salts that act as γ-phosphate mimics when combined with GDP. Based on the conformation of the active site, they conclude that GDP.BeF_3_- accurately mimics the GTP-bound state of tubulin, and GDP.AlF_3_- mimics a transitional GDP-Pi state. They show that the two fluoride salts both facilitate microtubule polymerization and stabilization, similar to the commonly used γ-phosphate mimic GMPCPP. They then use X-ray fiber diffraction to measure the lattice spacing of microtubules assembled with different nucleotides. Based on these measurements and the snapshots in the crystal structures, they present a model for microtubule polymerization.

Their key finding is that the lattice spacing in both the GTP-mimic state (GDP-BeF_3_-) and GDP-Pi-mimic state (GDP.AlF_3_-) is equivalent to that in the GDP state (~8.1nm/dimer). This contradicts the prevalent model in the field, in which the GTP-bound lattice has an extended spacing of ~8.3nm/dimer, and collapses to ~8.1nm/dimer upon hydrolysis and/or phosphate release. These conformational changes are thought to weaken lateral interactions, leading to depolymerisation being favoured in the GDP-bound state. As such, the results presented in this study challenge this model, and are such of great interest to the field. I would recommend this paper for publication, however I have some reservations with the manuscript, outlined below, that would require rectification prior to publication.

Major Points:

• The manuscript is currently lacking a direct comparison between the GMPCPP and GDP-BeF_3_- structures. In Figure 1B (other issues stated below), the nucleotide in the GMPCPP, GTP and BeF_3_- structures all appear to be in a comparable conformation. Given this, why would GMPCPP and BeF_3_- stabilized microtubules display different lattice compaction states? Following this, if the GMPCPP and BeF_3_- structures are equivalent, why should the BeF_3_- lattice spacing be taken to be the state of the GTP lattice rather than GMPCPP? The authors conclude that GMPCPP is represents a "hyper-active" transition state, however this point has not been argued, and is not intuitive to me.

• Figure 1: This figure is very unclear. In Figure 1B, there are labels for four different structures (GMPCPP, GDP, GTP, BeF_3_-), however only two of these structures are shown in the ribbon representation. Despite this, I can make out four nucleotides and magnesium ions. The nucleotides are all superimposed but coloured the same (for non-carbon atoms), so it is difficult to see differences between them. In Figure 1C the viewpoint is different to Figure 1B, despite them being images of the same site in the protein and comparing the same features. It is unclear why they do not show the GTP structure in Figure 1C but do in 1B. Residues and loops are described in the text (subsection “Phosphate analogues mimic activation and transition states at the hydrolytic site”) without being shown or labelled in the figure. The colours of the GMPCPP and AlF_3_- structures are too similar and should be changed. The authors should present this data more clearly and consistently. I think individual structures should be shown as well as comparisons, because as it is information is getting lost. Most notably, it is very difficult to see how the AlF_3_- ion is being coordinated given that everything else is overlapping.

• Subsection “Phosphate analogues reveal lattice features for the GTP/GDP•Pi-bound states”, paragraph three, Figure supplement – In the diffraction patterns where it is concluded that there is no 8nm reflection (e.g., GDP-MT, GDP-BeF_3_-MT), the position that would be occupied by the 8nm reflection is obscured by the central peak. This makes it impossible to judge whether this reflection is present or not. If the authors can provide diffraction patterns with the contrast/saturation adjusted so that these positions are visible against the background, this concern would be rectified. However, if the background at the position of the reflections is never low enough to see the reflections, any discussion regarding the (lack of an) 8nm reflection should be removed from the manuscript.

Reviewer #2:

Estevez-Gallego et al. have studied how different nucleotide analogs affect the conformation of ab-tubulin in its unpolymerized form (X-ray crystallography) or in microtubules (X-ray fiber diffraction). Based on their results they propose a model for the evolution of the stabilizing cap that differs from the dominant one, which is based primarily on structural observations from Eva Nogales' lab that assumed GMPCPP was a faithful mimic of GTP. There is much to like about the manuscript – I find the questions they are asking to be timely, and the proposal that the expanded state of the lattice is associated with π release is thought-provoking. I also think it is good for the field to have some uncertainty injected back into the discussion about the nature of the stabilizing cap and its relationship to nucleotide state. These are strengths of the manuscript. However, the manuscript also has significant weaknesses: the general presentation will in my opinion not be very accessible to a general reader, the fiber diffraction analysis is particularly hard to follow/assess, the new model for cap evolution is not well integrated with other findings, and it's not clear (to me) that the discussion of EB proteins in the context of this new model makes physical sense. Overall the negatives currently outweigh the positives.

Reviewer #3:

This work describes an extensive structural study on tubulin in complex with stathmin, and tubulin polymerized into microtubules, under various nucleotide conditions. Overall, this is a great piece of work that deserves consideration by *eLife*. However, I believe there are a few points that could make the paper more convincing with regard to their interpretations. The presented data is mostly of excellent quality and state of the art. Some parts I might not fully understand, which may be due to my own deficits, hence, if the authors can prove me wrong, that would be fine. However, below are a few points that I would like to be answered before publication.

The crystal structures are from stathmin-tubulin complexes where stathmin induces a bent conformation, no matter which nucleotide is bound. Obviously, X-ray crystallography is the authors line of work, and the data itself looks flawless to me, but some interpretations that have been extrapolated to pure microtubules might be critical. However, given the most recent successes with microtubules and high-resolution cryo-EM, reaching towards 3 Å resolution these days, it might be useful to include such data for completeness, and confirmation of the predicted models, which, as we all know, are just models, and not structures.

I am not sure where again this "expanded lattice" model came from initially, but I am somewhat skeptical about its real existence.… If there is really such an expansion as modelled in Figure 4, this should have been seen way before by simple cryo-EM, even without any sophisticated image reconstruction procedures. What we have seen in cryo-EM is that microtubules that were polymerized in vitro (and maybe even in vivo: see Chretien's work) occasionally incorporate, or loose a protofilament, making them go from e.g., 13's to 14's. With cryo-EM it would be easy to distinguish this process from an actual expansion process. Unfortunately, also the Zhang publications do not show any cryo-EM raw data, although an expansion should be seen on such pictures.

The authors should mention that not only GTP hydrolysis affects binding factors via changes in the tubulin lattice, but more likely, binding factors change the tubulin lattice and strongly stabilize it (e.g., tau, some kinesins, etc.).

It is not clear to me how the authors measured the percentage of different types of MTs (e.g., 12-pfs versus 14ps, etc.) in their solutions. The supplementary figure shows very similar supertwist (and overall) patterns in all states (see also my comment on figure 4).

Figure 1: As I understand, these structures come from tubulin-stathmin (plus TTL) complexes. Hence, the conformations found there could be affected by that situation, and be quite different from a real microtubule complex. Otherwise, the quality of the data is impressive.

Figure 2: The negative stain electron micrographs are horrible (sorry)… showing huge amounts of junk next to the tubes plus vast amounts of depolymerized oligomers, especially in B. The little oligomers are formed after rapid depolymerization, and are often observed, but the large bits of junk might be caused by some purification issues, or else.

Figure 3: I must admit that, despite having dealt extensively with microtubule layerline patterns, this figure is a bit a mystery to me, especially the graphs D and E. Also, I cannot follow the protofilament estimation on 3C. However, the layerline pattern appears to come mostly from supertwisted microtubules that are larger in protofilament size than 13-type microtubule (the outer densities in layerline 1 of the supplementary data are all slightly beyond layerline 1, indicating a right-handed supertwist). This is not surprising given the polymerization conditions used (e.g., GMP-CPP produces mostly 14-protofilament MTs), and it does not really affect the interpretation on the various GTP states, except that the outer densities at layerline 1 indicates a supertwist, which should be noted. Also, typically layerline 1 should be assigned to the 8nm repeat (The tubulin-dimer length). Since that layerline is well visible in all diffractions, the numbering should start there. Otherwise, marking the 4-nm line as #1 would suggest that α- and β-tubulin are identical. In principal, with supertwisted microtubules as the authors present, the layerline numbering should start with the supertwist helix (which is not visible here, except for the outer densities in layerline 1), and then, layerline 1 in this figure would actually be 34 or something.

Figure 4: The model of the expanded microtubule is mostly wrong. The seam, if present, is not including such a strange rearrangement of dimers as drawn here. The lateral contacts at the seam in this section are wrong as well and appears to go from one tubulin monomer to the interaction site of the next lateral dimer. Furthermore, the model of an expanded lattice, how do the authors know if this is not an effect of larger protofilament numbers at these nucleotide states (e.g., 14's instead of 13's)? To clarify this the authors would have to do a more carefully cryo-EM study. This way, the appearance of supertwisted microtubules becomes obvious directly from the micrographs. Cryo-EM of microtubules, if one does not aim for near-atomic resolution, is quite straightforward.

The supplementary figure compares diffraction from all these states. However, I have looked at helical tubulin diffraction for a good part of my scientific life, and I cannot see any major differences from that figure. Either there are none, or they should be made more obvious. Also, if some states should make a majority of 12-protofilament microtubules (e.g., with BeF_3_), then the supertwist reflection in what is here wrongly layerline 1, should be inside (towards the center of the diffraction), not outside as this reflection is in all shown diffractions.

Conclusions: If the authors can convince me of their points and answer my questions, the paper could be published.

[Editors’ note: further revisions were suggested prior to acceptance, as described below.]

Thank you for submitting your article "Structural model for differential cap maturation at growing microtubule ends" for consideration by *eLife*. Your article has been reviewed by two peer reviewers, and the evaluation has been overseen by a Reviewing Editor and Cynthia Wolberger as the Senior Editor. The following individual involved in review of your submission has agreed to reveal their identity: Andreas Hoenger (Reviewer #3).

The reviewers have discussed the reviews with one another and the Reviewing Editor has drafted this decision to help you prepare a revised submission.

Summary:

The manuscript by Estévez-Gallego et al. sets out to investigate the conformational state of tubulin in different nucleotide states, aiming to shed light on the mechanism of dynamic instability in microtubules. They present crystal structures of curved tubulin dimers in the presence of two fluoride salts that act as γ-phosphate mimics when combined with GDP. Based on the conformation of the active site, they conclude that GDP-BeF_3_- accurately mimics the GTP-bound state of tubulin, and GDP-AlF_3_- mimics a transitional GDP-Pi state. They show that the two fluoride salts both facilitate microtubule polymerization and stabilization, similar to the commonly used γ-phosphate mimic GMPCPP. They then use X-ray fiber diffraction and cryoEM imaging to measure the lattice spacing of microtubules assembled with different nucleotides. Based on these measurements and the snapshots in the crystal structures, they present a model for microtubule polymerization.

Their key finding is that the lattice spacing in both the GTP-mimic state (GDP-BeF_3_-) and GDP-Pi-mimic state (GDP-AlF_3_-) is equivalent to that in the GDP state (~8.1nm/dimer). This contradicts the prevalent model in the field, in which the GTP-bound lattice has an extended spacing of ~8.3nm/dimer, and collapses to ~8.1nm/dimer upon hydrolysis and/or phosphate release. These conformational changes are thought to weaken lateral interactions, leading to depolymerisation being favoured in the GDP-bound state. As such, the results presented in this study challenge this model, and are such of great interest to the field.

Essential revisions:

Reviewer #2:

The authors have done a nice job responding to the initial reviews, both in terms of making the writing clearer and also by incorporating cryo-EM data to complement the fiber diffraction.

I have one comment about the presentation that I think it would be nice for the authors to address more clearly. While the authors data do seem to support the idea that the conformation of tubulin in the GTP lattice is 'compacted' (not expanded as seen with GMPCPP), the way the manuscript is written a reader might get the impression that the expanded conformation is entirely an artifact of using GMPCPP. While the authors mention in passing that an expanded conformation has been observed for yeast tubulin, I think it would be good if they also mentioned studies of *S. pombe*microtubules (from Carolyn Moores) and *C. elegans*microtubules (from Gary Brouhard), both of which also showed expanded lattices in non-GMPCPP states. I think a more nuanced discussion of this and what the authors think it might mean for dynamics and EB recognition would strengthen the manuscript.

Reviewer #3:

This paper, overall, improved substantially. It seems as if a whole bunch of additional data has been produced, including the addition of the lattice analysis by Denis Crétien.

I still think the state of the negative staining EM in Figure 2 B and F are not very useful, and too small.

Figure 4A would benefit significantly from displaying what is there, but a bit larger and maybe shorter, and with an added axially compressed image that emphasizes the supertwist features, which here, for people only partially involved in this are invisible.

Also, the relationship between A and D is not directly visible. It could be worked out with a better display of the diffraction data. Left vrs right-handed supertwists have their distinct patterns.

Figure 4B is not helpful in this. The supertwist should show up as two strong layerlines around Bessel order -2, and the pattern there is indicative for left or right handedness.

---

## [Author Response]

[Editors’ note: The authors appealed the original decision. What follows is the authors’ response to the first round of review.]

Reviewer #1:[…] Major Points:• The manuscript is currently lacking a direct comparison between the GMPCPP and GDP-BeF_3_- structures. In Figure 1B (other issues stated below), the nucleotide in the GMPCPP, GTP and BeF_3_- structures all appear to be in a comparable conformation. Given this, why would GMPCPP and BeF_3_- stabilized microtubules display different lattice compaction states?

As mentioned by the reviewer, there are no differences when comparing high-resolution tubulin bent structures of the BeF_3_ -, GTP or GMPCPP-bound nucleotides. All three structures show the γ-phosphates/ BeF_3_ – and T5 loops in very close positions denoting an equivalent β-tubulin top-surface prone to assembly. Hence, we presume that major differences occur on the straight conformation. Interestingly, there is a known linear dependence of tubulin assembly and Mg^2+^, which is incorporated into the lattice. However, from published cryo-EM MT structures, only CPP-MTs show this ion at the nucleotide-binding site. This could be related to a wash out effect during sample preparation or to a decrease on the occupancy due to weaker coordination. This difference would make questionable any comparison between curved (high-resolution and Mg^2+^) vs. straight (lower-resolution and no Mg^2+^) structures.

Following this, if the GMPCPP and BeF_3_- structures are equivalent, why should the BeF_3_- lattice spacing be taken to be the state of the GTP lattice rather than GMPCPP?

There are several lines of evidence indicating that lattice expansion is not related to the GTP-bound state. First, GMPPCP-MTs (another GTP analogue) display a compact state similarly to BeF_3_-. Second, GMPCP-MTs (a GDP analogue and the hydrolyzed version of GMPCPP) show an expanded lattice, suggesting a link between the presence of the methylene group in these analogues and the lattice expansion. It should be noticed that ATP analogues chemical properties are quite different than those of ATP, they display phosphate ionization constants (Blackburn GM, Kent DE, Kolkmann F. 1981. JCS Chem Comm: 1188-1190) and bind divalent ions with highly different strengths (Yount RG, Babcock D, Ballantyne Wm, Ojala D. 1971. Biochemistry 10 (13): 2484-89). This is key and makes the difference so AMP-PNP (an ATP analogue) can be a substrate or cannot be for different ATPases (Yount RG, Babcock D, Ballantyne Wm, Ojala D. 1971. Biochemistry 10 (13): 2484-89). Hence, we presume that the insertion of the methylene group between α and β phosphate might induce such differences at the nucleotide properties that favors such huge differences on the protein-protein interface. Which are the specific changes on the nucleotides, and how these are affecting tubulin axial interface on the straight conformation is difficult to figure out from available structures because the unambiguous determination of nucleotides geometry requires resolutions below 3.5Å. We have now included this comparison on the Results section (subsection “MT lattice expansion is not related to its GTP-bound state”).

The authors conclude that GMPCPP is represents a "hyper-active" transition state, however this point has not been argued, and is not intuitive to me.

We apologize for the misunderstanding. We defined the expanded lattice as a ‘hyper-active’ state because this structure is linked to highly stable MTs (GMPCPP and paclitaxel). However, we agree that this is inexact because GMPCP does not favor tubulin nucleation whereas it induces expanded lattices. Thus, we have avoided the use of this term in the new version of the manuscript.

• Figure 1: This figure is very unclear. In Figure 1B, there are labels for four different structures (GMPCPP, GDP, GTP, BeF_3_-), however only two of these structures are shown in the ribbon representation. Despite this, I can make out four nucleotides and magnesium ions. The nucleotides are all superimposed but coloured the same (for non-carbon atoms), so it is difficult to see differences between them. In Figure 1C the viewpoint is different to Figure 1B, despite them being images of the same site in the protein and comparing the same features. It is unclear why they do not show the GTP structure in Figure 1C but do in 1B. Residues and loops are described in the text (subsection “Phosphate analogues mimic activation and transition states at the hydrolytic site”) without being shown or labelled in the figure. The colours of the GMPCPP and AlF_3_- structures are too similar and should be changed. The authors should present this data more clearly and consistently. I think individual structures should be shown as well as comparisons, because as it is information is getting lost. Most notably, it is very difficult to see how the AlF_3_- ion is being coordinated given that everything else is overlapping.

We have followed the reviewers’ advice in making a new figure. In one panel we highlight the residues mentioned on the manuscript (as stick representations) and secondary structural elements surrounding the nucleotide. In several panels we display the superposition of BeF_3_ – and AlF_3_ with other nucleotide bound structures. We have decreased the intensity of GMPCPP orange to increase the differences between GMPCPP and AlF_3_. These panels display the same orientation to make them clearer for the reader. We also have added the composite omit map of each of the structures solved, including two views of AlF_3_ so the reader can have a better overview of the AlF_3_ position and coordination. We hope that the figure is clearer for the reader.

• Subsection “Phosphate analogues reveal lattice features for the GTP/GDP•Pi-bound states”, paragraph three, Figure supplement – In the diffraction patterns where it is concluded that there is no 8nm reflection (e.g., GDP-MT, GDP-BeF_3_-MT), the position that would be occupied by the 8nm reflection is obscured by the central peak. This makes it impossible to judge whether this reflection is present or not. If the authors can provide diffraction patterns with the contrast/saturation adjusted so that these positions are visible against the background, this concern would be rectified. However, if the background at the position of the reflections is never low enough to see the reflections, any discussion regarding the (lack of an) 8nm reflection should be removed from the manuscript.

We agree that it is difficult to see the 8nm-1 reflection in the figure presented. This is because under some conditions MT nucleation is defective whereas MT elongation is favored producing fewer but longer MTs, which has a strong effect of the resulting diffracting image and forced us to increase brightness/contrast levels to show the resolution of meridional signals (1nm-1 layer line). This is detrimental to equatorial signals that become very wide. We have prepared a new supplementary figure (Figure 3—figure supplement 2) where each condition is represented by one image showing the meridional signals and an inset image displaying the equatorial signals.

Reviewer #2:Estevez-Gallego et al. have studied how different nucleotide analogs affect the conformation of ab-tubulin in its unpolymerized form (X-ray crystallography) or in microtubules (X-ray fiber diffraction). Based on their results they propose a model for the evolution of the stabilizing cap that differs from the dominant one, which is based primarily on structural observations from Eva Nogales' lab that assumed GMPCPP was a faithful mimic of GTP. There is much to like about the manuscript – I find the questions they are asking to be timely, and the proposal that the expanded state of the lattice is associated with π release is thought-provoking. I also think it is good for the field to have some uncertainty injected back into the discussion about the nature of the stabilizing cap and its relationship to nucleotide state. These are strengths of the manuscript. However, the manuscript also has significant weaknesses: the general presentation will in my opinion not be very accessible to a general reader, the fiber diffraction analysis is particularly hard to follow/assess, the new model for cap evolution is not well integrated with other findings, and it's not clear (to me) that the discussion of EB proteins in the context of this new model makes physical sense. Overall the negatives currently outweigh the positives.

We acknowledge the interest expressed for the manuscript. To improve it we have worked on making easier to follow the fiber diffraction analysis, the reasoning of the model and its integration with other findings. We have done our best to simplify the results obtained.

Reviewer #3:This work describes an extensive structural study on tubulin in complex with stathmin, and tubulin polymerized into microtubules, under various nucleotide conditions. Overall, this is a great piece of work that deserves consideration by eLife. However, I believe there are a few points that could make the paper more convincing with regard to their interpretations. The presented data is mostly of excellent quality and state of the art. Some parts I might not fully understand, which may be due to my own deficits, hence, if the authors can prove me wrong, that would be fine. However, below are a few points that I would like to be answered before publication.The crystal structures are from stathmin-tubulin complexes where stathmin induces a bent conformation, no matter which nucleotide is bound. Obviously, X-ray crystallography is the authors line of work, and the data itself looks flawless to me, but some interpretations that have been extrapolated to pure microtubules might be critical. However, given the most recent successes with microtubules and high-resolution cryo-EM, reaching towards 3 Å resolution these days, it might be useful to include such data for completeness, and confirmation of the predicted models, which, as we all know, are just models, and not structures.

In this work we aimed to get an insight into the activation and stabilization mechanism on tubulin/MTs, and to this end we have tested a large number of conditions. As mentioned to reviewer 2, we used fiber diffraction because this technique allows:

1) The inspection of filamentous samples obtained under a wide range of tested conditions, some of which would entail technical issues difficult to get over with cryo-EM (i.e. the use of crowding agents such as glycerol and methylcellulose that poorer S/N ratio).

2) A quick data acquisition with a good throughput and accumulation of data from native (not frozen) samples obtained under a wide range of different conditions (i.e. solving the issue of comparability between tested assembly conditions).

3) Very accurate measurements of both overall MT dimension and axial spacing because readily distinguishing between features that are identical in the population (monomer and dimer rise) from those that are not (PF number, supertwist).

4) Getting a much higher S/N ratio for the equatorial signals (i.e. reflecting lateral interactions) that are clear at the J04 + Jn1 (giving good estimations of the mean diameter of MTs as well as the spacing between PFs (i.e., PF number)).

We totally agree that Cryo-EM structures will contribute on understanding the nature of protein protein interaction within the lattice but disagree on the resolution achieved. Most of the structures are far from the aforementioned 3 Å (some of them are at 3.5 Å but most of them are in the 4.2-4.5Å range). And high-resolution is crucial at some stages. For instance, we have argued to reviewer 2: ‘in the view of our results we speculate that the GDP-Pi-MTs from Manka and Moores, 2018 are actually GTP-MTs’. It is easy to understand that at 4.2Å and in the view of the general consensus about GTP-bound state that must be in an expanded conformation, they assumed they got GDP-Pi instead of GTP at the nucleotide-binding site. However, both their map (that shows a continuous density with no breaks) and their PDB file (that shows a π at bonding distance of the β-phosphate) cast some doubts on their interpretation.

Considering the common voice from all three reviewers claiming for a direct validation of our fiber diffraction data with cryo-EM image, we have built collaboration with Denis Chretien (now a co-author in this work). He has contributed with cryo-EM images of MTs, their FFT and the analysis of MTs subpopulations according to the PF number under GDP-, GMPCPP- and BeF_3_ -conditions. We are really happy to show that these data correlate very well with previously presented x-ray shear-flow aligned fiber diffraction analyses and we hope the results presented are more convincing now.

*I am not sure where again this "expanded lattice" model came from initially, but I am somewhat skeptical about its real existence… If there is really such an expansion as modelled in Figure 4, this should have been seen way before by simple cryo-EM, even without any sophisticated image reconstruction procedures. What we have seen in cryo-EM is that microtubules that were polymerized* in vitro *(and maybe even* in vivo*: see Chretien's work) occasionally incorporate, or loose a protofilament, making them go from e.g., 13's to 14's. With cryo-EM it would be easy to distinguish this process from an actual expansion process. Unfortunately, also the Zhang publications do not show any cryo-EM raw data, although an expansion should be seen on such pictures.*

This is an important point and we have had extensive discussions with Denis Chrétien. Now we first discuss the model that is directly derived from our data: tubulin-GTP (BeF_3_ -) goes to GDPPi state (AlF_3_) and GDP-state without any change in the longitudinal dimension. This already challenges the established model of an extended GTP-state (GMPCPP), which would compact upon GTP hydrolysis (GTP-γ-S) proposed by the Nogales lab. Our cryo-EM data also show that the rotations involved in this model are simple consequences of the lattice accommodation to subtle changes in tubulin dimensions or helical rise.

However as a speculation based on indirect data, we now hypothesize that tubulin could expand after hydrolysis to facilitate π release (in terms of a putative Pi-Mg^2+^ complex release or charge compensation in the absence of Mg^2+^ coordination as presented in actual cryo-EM structures). It is true that nobody has observed this before by cryo-EM (including DC) implying that if expansion exits, it happens in a very narrow region (i.e. a few tens of nm) and/or it is a random process (making it blurred in the MT lattice). We hope that presenting this provocative hypothesis will force cryo-EM people to have a closer look at growing MT ends. We also hope that the new technologies available today (high-end cryo-EMs, direct-electron detectors) will allow cryo-EM specialist to challenge our hypothesis.

The authors should mention that not only GTP hydrolysis affects binding factors via changes in the tubulin lattice, but more likely, binding factors change the tubulin lattice and strongly stabilize it (e.g., tau, some kinesins, etc.).

True. Many of these partner proteins actually modify MTs dynamics so it is not a big surprise that they modify the lattice structure. In fact, we have seen that CP-MTs show an expanded lattice whereas in Zhang et al., 2018 they found CP-MT-EB3 in a compact state. We suggest that the compaction might be a consequence of EB binding (Discussion paragraph five).

It is not clear to me how the authors measured the percentage of different types of MTs (e.g., 12-pfs versus 14ps, etc.) in their solutions. The supplementary figure shows very similar supertwist (and overall) patterns in all states (see also my comment on Figure 4).

We have prepared a Figure 4—figure supplement 1 and improved the Materials and Methods section to make it clear to a general reader. As you indicated, the layer lines of 4nm-1 of tubulin repeat are curved partially due to the super-twisting of 14-15 MTs as described by Chretien and Fuller (Chretien and Fuller, 2000), but we did not use this profile data for further analysis because our methods could not completely exclude the possibility of MT misalignment by shearing (we estimated ca. 2-3º of deviations, paragraph two subsection “Phosphate analogues reveal lattice features for the GTP/GDP•Pi-bound states”). Curved layer line coming from misaligned MTs is more obvious in 1nm-1 signal. It also depended on the mechanical properties (flexural rigidity or persistence length) of MTs. We could not control such parameters. Thus, we suppose the J04 + Jn1 signal profile should be the best for the estimation of PF numbers. The correlation between these and cryo-EM data supports our calculations.

Figure 1: As I understand, these structures come from tubulin-stathmin (plus TTL) complexes. Hence, the conformations found there could be affected by that situation, and be quite different from a real microtubule complex. Otherwise, the quality of the data is impressive.

We thank the reviewer for the comment. We agree that in MTs (straight conformation) the picture can be different. Unfortunately, cryo-EM structures are not giving enough resolution to build an unambiguous geometry of the nucleotides either (see comment above) making the full comparison difficult to approach

Figure 2: The negative stain electron micrographs are horrible (sorry)… showing huge amounts of junk next to the tubes plus vast amounts of depolymerized oligomers, especially in B. The little oligomers are formed after rapid depolymerization, and are often observed, but the large bits of junk might be caused by some purification issues, or else.

We were aware about that! Although we have no clear cryo-EM images, we decided to keep the negative stain micrographs because we feel that the images are of sufficient quality to show that the change of turbidity in the Figure is due to MT polymerization and not another type of aggregation (i.e., tubulin rings formation).

Unfortunately it is impossible to get better images because most of the junk comes from BeF_3_ – and AlFx precipitation (likely due to their high concentrations in the acidic environment of the uranyl acetate staining solution). Furthermore, in our experimental solutions the amount of protein is definitely huge for EM experiments which further complicate to get better images.

Figure 3: I must admit that, despite having dealt extensively with microtubule layerline patterns, this figure is a bit a mystery to me… especially the graphs D and E. Also, I cannot follow the protofilament estimation on 3C. However, the layerline pattern appears to come mostly from supertwisted microtubules that are larger in protofilament size than 13-type microtubule (the outer densities in layerline 1 of the supplementary data are all slightly beyond layerline 1, indicating a right-handed supertwist). This is not surprising given the polymerization conditions used (e.g., GMP-CPP produces mostly 14-protofilament MTs), and it does not really affect the interpretation on the various GTP states, except that the outer densities at layerline 1 indicates a supertwist, which should be noted. Also, typically layerline 1 should be assigned to the 8nm repeat (The tubulin-dimer length). Since that layerline is well visible in all diffractions, the numbering should start there. Otherwise, marking the 4-nm line as #1 would suggest that α- and β-tubulin are identical. In principal, with supertwisted microtubules as the authors present, the layerline numbering should start with the supertwist helix (which is not visible here, except for the outer densities in layerline 1), and then, layerline 1 in this figure would actually be 34 or something.

Panels 3D and 3E are the Intensity profiles vs. distances. We apologize the panel 3C is difficult to follow (reviewer 2 also pointed out this) and we have changed this plot into bars representation. We have improved this figure and prepared a supplementary figure to show our pipeline of work and data analysis. Hope this make the message clearer.

We have named the 4nm-1 repeat as the first layer line according to previous publications (Kamimura et al., 2016); we apologize because this is not consistent with the nomenclature in the EM field.

Figure 4: The model of the expanded microtubule is mostly wrong. The seam, if present, is not including such a strange rearrangement of dimers as drawn here. The lateral contacts at the seam in this section are wrong as well and appears to go from one tubulin monomer to the interaction site of the next lateral dimer.

We apologize for the lack of consistency on the scheme. Lattice expansion was significantly emphasized to make easy to follow for a general reader. We have worked on it to fit the seam issues.

Furthermore, the model of an expanded lattice, how do the authors know if this is not an effect of larger protofilament numbers at these nucleotide states (e.g., 14's instead of 13's)? To clarify this the authors would have to do a more carefully cryo-EM study. This way, the appearance of supertwisted microtubules becomes obvious directly from the micrographs. Cryo-EM of microtubules, if one does not aim for near-atomic resolution, is quite straightforward.

We can rule out this possibility because fiber diffraction allows very accurate measurements and readily distinguishing between features that are identical in the population (monomer and dimer rise) from those that are not (diameter, PF number). The PF number do not infer on the axial spacing estimation as probed by Px-MTs (expanded lattice) have an average PF number of 12.37 and AlFx-MTs (compact lattice) contain an average PF number of 13.4.

However, following the reviewers’ general claim, we have included cryo-EM images of GDP-, GMPCPP- and BeF_3_-MTs. And these images beautifully correlate with the fiber diffraction data.

The supplementary figure compares diffraction from all these states. However, I have looked at helical tubulin diffraction for a good part of my scientific life, and I cannot see any major differences from that figure. Either there are none, or they should be made more obvious. Also, if some states should make a majority of 12-protofilament microtubules (e.g., with BeF_3_), then the supertwist reflection in what is here wrongly layerline 1, should be inside (towards the center of the diffraction), not outside as this reflection is in all shown diffractions.

Indeed, in 12_3 microtubules (12 protofilaments and 3-start monomer helices), the outer densities on the ‘4 nm layer line’ are away from the equator since the protofilaments are righthanded. This is now displayed in Figure 4—figure supplement 1, which compares diffraction patterns of 12_3 and 14_3 microtubules.

Conclusions: If the authors can convince me of their points and answer my questions, the paper could be published.

We hope our comments sound convincing and changes on the manuscript satisfy the reviewer’s expectations.

[Editors’ note: what follows is the authors’ response to the second round of review.]

Essential revisions:Reviewer #2:The authors have done a nice job responding to the initial reviews, both in terms of making the writing clearer and also by incorporating cryo-EM data to complement the fiber diffraction.I have one comment about the presentation that I think it would be nice for the authors to address more clearly. While the authors data do seem to support the idea that the conformation of tubulin in the GTP lattice is 'compacted' (not expanded as seen with GMPCPP), the way the manuscript is written a reader might get the impression that the expanded conformation is entirely an artifact of using GMPCPP. While the authors mention in passing that an expanded conformation has been observed for yeast tubulin, I think it would be good if they also mentioned studies of S. pombe microtubules (from Carolyn Moores) and C. elegans microtubules (from Gary Brouhard), both of which also showed expanded lattices in non-GMPCPP states. I think a more nuanced discussion of this and what the authors think it might mean for dynamics and EB recognition would strengthen the manuscript.

We agree, actually we think that it is very likely that the expanded conformation has any physiological relevance; accordingly we have followed the reviewer advice and made some modification among the Discussion section. However, more high-resolution data of those MTs (yeast and worm) on the GTP-bound state will be key on unraveling the compact-to-expanded transition in non-mammalian MTs

Reviewer #3:This paper, overall, improved substantially. It seems as if a whole bunch of additional data has been produced, including the addition of the lattice analysis by Denis Crétien.I still think the state of the negative staining EM in Figure 2B and F are not very useful, and too small.

In this point we disagree because we feel that it is still important to illustrate how length measurements were performed, and also assess the presence of MTs in the sample. However accordingly to the recommendations we have moved it to Figure 2—figure supplement 1.

Figure 4A would benefit significantly from displaying what is there, but a bit larger and maybe shorter, and with a added axially compressed image that emphasizes the supertwist features, which here, for people only partially involved in this are invisible.

We are not keen on doing an anisotropic reduction of size of these images, since we feel that it may confuse the reader. We are aware that the moiré pattern in 13_3 GDP-BeF_3_^-^- and GMPCPP-MTs is indeed very long, which reflects the fact that their PF skew angle is very low (-0.20° and +0.30° respectively for the MTs displayed in the figure), but we think that an axial reduction in size would give the impression that the protofilament skew angle is higher than those values. In order to help the reader, we have added numbers with arrows close to the filtered images, which indicate the number of internal dark fringes inside the images and how they are off-center with respect to the microtubule axis. We have modified the figure legend accordingly.

Also, the relationship between A and D is not directly visible.

Following the advice we have modified panel A. The relationship is the following: when fringes are parallel (GDP-MT in A) the protofilaments are not skewed (θ=0° in D), and when they show moiré patterns (GDP-BeF_3_^-^- and GMPCPP-MTs in A), the protofilaments are skewed (θ = -0.20° or +0.30°, respectively, in D). We hope that this relationship is now more straightforward visible.

It could be worked out with a better display of the diffraction data. Left vrs right-handed supertwists have their distinct patterns.

In order to better display the data the supertwist is now shown in the blow-up images of Figure 4C. In this way Figure 4B highlight the differences in tubulin monomer spacing along PFs between the different types of MTs, i.e. the relative positions of their nominal ‘4 nm layer lines’. Only these values are given in this panel.

Figure 4B is not helpful in this. The supertwist should show up as two strong layerlines around Bessel order -2, and the pattern there is indicative for left or right handedness.

We respectfully disagree, the separation between the *J_S_* and *J_N_S_* layer lines will depend directly on the protofilament skew angle. Since the angle is very low in 13_3 GDP-BeF_3_^-^- and GMPCPP-MTs (see above), they remain very close to each other in their diffraction patterns, and thus do not appear as “two strong layer lines”. However, their respective position with respect to the equator provides a direct measure of the PF handedness (see legend). Note that when the protofilament skew increases such as in 12_3 or 14-MTs (Figure 4—figure supplement 1), the separation between those layer lines also increases, but they still remain close to each other. Our experience with more skewed MT types such as 15_3, 15_4, 14_4 or 13_4 tells us that to have two strong and well separated *J_S_* and *J_N_S_* layer lines as suggested by the reviewer, but one needs PF skew angles larger than ~ ± 1.5° to observe this pattern, which is not the case for the MT types reported in this study.